# Prevalence and associated factors of playing-related musculoskeletal disorders among music students in Europe. Baseline findings from the Risk of Music Students (RISMUS) longitudinal multicentre study

Cinzia Cruder [1,2,3]*, Marco Barbero[1], Pelagia Koufaki[2], Emiliano Soldini[4], Nigel Gleeson[2]

1 Department of Business Economics, Rehabilitation Research Laboratory 2rLab, Health and Social Care, University of Applied Sciences and Arts of Southern Switzerland, Manno, Switzerland, 2 Centre for Health, Activity and Rehabilitation Research, Queen Margaret University, Edinburgh, United Kingdom, 3 Department of Research and Development, Conservatory of Southern Switzerland, Lugano, Switzerland, 4 Department of Business, Research Methodology Competence Centre, Health and Social Care, University of Applied Sciences and Arts of Southern Switzerland (SUPSI), Manno, Switzerland

* cinzia.cruder@supsi.ch, Ccruder@qmu.ac.uk

**Data Availability Statement:** Data cannot be shared publicly because they contain potentially

## Abstract

Musculoskeletal (MSK) conditions among professional musicians and music students are frequent and may have significant physical and psychosocial consequences on their lives and/or on their playing abilities. The Risk of Music Students (RISMUS) research project was set up in 2018 to longitudinally identify factors associated with increased risk of playing-related musculoskeletal disorders (PRMDs) in a large sample of music students enrolled in pan-European institutions. The aim of this cross-sectional study was to describe the prevalence of playing-related musculoskeletal disorders (PRMDs) in this novel population at baseline of the RISMUS project. A further goal was to begin to identify variables that might be associated with the self-reported presence of PRMDs among music students. Eight hundred and fifty students from fifty-six conservatories and music universities in Europe completed a web-based questionnaire on lifestyle and physical activity participation levels, musical practice habits, health history and PRMDs, psychological distress, perfectionism and fatigue. A total of 560 (65%) out of 850 participants self-reported a positive history of painful MSK conditions in the last 12 months, 408 (48%) of whom self-reported PRMDs. Results showed that coming from West Europe, being a first- or a second-year Masters student, having more years of experience and higher rates of perceived exertion after 45 minutes of practice without breaks were factors significantly associated with self-reported presence of PRMDs. According to the authors' knowledge, a large-scale multicentre study investigating prevalence and associated factors for PRMDs among music students at different stages of their education (from Pre-college to Masters levels) has not been conducted before. The high prevalence of PRMDs among music students, especially those studying at university-level, has been confirmed in this study and associated factors have been

identifying or sensitive participants' information and disclosure to third parties has been prohibited by the QMU Ethics Committee. Data are only available for researchers who meet the criteria for access to confidential data and are stored at a secure server hosted by Queen Margaret University. These data can be made available to interested researchers upon request to the corresponding author, who will have to ask the permission for data access to the QMU Ethics Committee at ResearchEthics@qmu.ac.uk.

**Funding:** The research data reported in this article is part of RISMUS: A longitudinal investigation of the factors associated with increased RISk of playing-related musculoskeletal disorders in Music students, an investigation funded by the Swiss National Science Foundation (grant ref. 10531C_182226) and supported by Queen Margaret University-Edinburgh for the fulfilment of a PhD research award. This funding source had no role in the design of this study and will not have any role during its execution, analysis and interpretation of the data.

**Competing interests:** The authors have declared that no competing interests exist.

identified, highlighting the need for relevant targeted interventions as well as effective prevention and treatment strategies.

## Introduction

Musculoskeletal (MSK) conditions are a common concern in the general population and the most prevalent cause of serious, long-term pain and physical disability, affecting 25% of all adults across European countries [1–4]. Besides MSK conditions leading to physical and work disability, some occupational groups have higher prevalence of MSK conditions that may be caused by the nature of their work. Jobs with frequently repeated movements and high physical demands in combination with psychosocial stress symptoms are often associated with MSK conditions [5–8]. In this regard, musicians represent a profession associated with MSK and psychosocial demands [9,10] that may limit their physical abilities, having a significant impact on their performances [11,12] but also a marked effect on their lives [9,10,13,14].

The term playing-related musculoskeletal disorders (PRMDs) was introduced by Zaza et al. in 1998 ("any pain, weakness, numbness, tingling, or other symptoms that interfere with your ability to play your instrument at the level you are accustomed to") to identify musculoskeletal symptoms that interfere with the ability to play the instrument [12]. The most frequently reported factors common to the development of PRMDs in musicians include among others: the type of instrument, long hours of practice and insufficient rest breaks, poor physical condition, as well as muscle fatigue and overuse [13,15–17]. Furthermore, several studies revealed that there were positive associations between the presence of MSK conditions and psychological stressors (i.e. anxiety and stress, depression and perfectionism) [15,18–20].

Although the definition of PRMDs does not provide a causality of the disorder (i.e. the disorder is the result of playing the instrument), distinguishing PRMDs from non-playing-related or generic MSK conditions has the advantage of excluding symptoms without a significant impact and therefore not relevant for the musician [13,21]. Nevertheless, there are some discrepancies between terms that describe musicians' conditions in the relevant literature. Several studies investigated painful MSK conditions using other descriptions than PRMDs [21–25] or evaluated PRMDs without strictly following the definition or without reporting the definition in the methods section [26–29].

A recent systematic review has reported the point prevalence of MSK conditions among musicians as between 57 and 68% for all complaints, and between 9 and 68% for playing-related complaints; similarly, PRMD lifetime prevalence oscillated between 62% and 93% [30]. However, the variety of definitions and the heterogeneity of types of prevalence, as well as heterogeneity amongst study populations has made comparison of the data unviable in this systematic review [30]. For this reason, recent studies and reviews strongly recommended conducting future research regarding the epidemiology of musicians' PRMDs among large sample sizes, including the description of the measured MSK condition (i.e. distinguishing PRMDs and non-playing-related) and the use of adequate and validated outcome measures [18,30,31]. Similarly, the contemporary literature offers a large heterogeneity of methods amongst small samples that limit generalisations and meta-analytical synthesis of the evidence of music students' MSK conditions [30,46]. This is despite a growing literature regarding MSK among music students [10,11,25,28,32–39] and a proliferation of preventive courses as well as short-term health education programs during the last twenty years [40–45]. Furthermore, in contrast to the literature on MSK conditions in the general population, scientific evidence is scarce concerning prevalence rates and associated factors in subgroups of age and different

stages of occupation [30]. Therefore, there is a need to deliver robust and large-scale data on music students at different levels of their education to enhance an epidemiological appreciation of how best to prioritise the strategies for improving the management of PRMDs and to enhance evidence relating to the associated factors that may increase the risk of adverse outcomes.

The Risk of Music Students (RISMUS) research project was set up in 2018 to characterise clinical features of a large sample of students from pan-European music institutions and to longitudinally identify factors associated with increased and evolving risk of playing-related musculoskeletal disorders during their professional training [47]. This 12-months longitudinal multicentre investigation has evolved to incorporate recommendations within the current literature [18,30,31,46] that an effective way in which to predict the occurrence of PRMDs among musicians would be to conduct a longitudinal study, using an online-based administration of questionnaires. While necessarily involving musicians' self-perception of status, it nevertheless has the benefit of being able to reach a larger population sample, addressing an identified gap in the existing literature.

### Aims

The purpose of the present study was to examine the prevalence of PRMDs in a large-scale study population of music students enrolled in different pan-European music institutions at baseline of the RISMUS project, in order to characterise the study population at different levels of training (i.e. university-level students and Pre-college students). Our hypothesis was that there is a higher prevalence of PRMDs among university-level students in comparison with Pre-college students (i.e. transition between Pre-college and university-level) possibly due to the assumption that the exposure of playing-related activities is progressively demanding throughout their training. A further goal was to begin to identify variables that might be associated with the self-reported presence of PRMDs among music students. Specifically, an approach involving multivariable modelling might offer preliminary explorative and novel insights of the baseline findings to be further verified within the longitudinal analyses.

## Materials and methods

This cross-sectional study focuses on part of the baseline data of RISMUS and refers to the overview of data from all music students participating in the research. One hundred and ninety schools have been invited to participate in this research and fifty-six of the approached institutions accepted to take part and contributed to the recruitment, by distributing the link to a web-based questionnaire to their student groups. The web-based questionnaire included questions about any PRMD that students had experienced and different potential risk factors. Before starting any procedure, participants had to complete and sign an electronic written consent form. Although beyond the scope of the present article, future articles will disseminate follow-up data from RISMUS considering PRMD aetiology amongst the professional training of music students. The research project was granted ethical approval by the Research Ethics Committee of Queen Margaret University of Edinburgh (REP 0177).

### Participants

A total of nine hundred and ninety-seven students were recruited from the school registries of the aforementioned schools (see Table 1) for the baseline data collection between November 2018 and January 2019.

Inclusion criteria were men and women over 18 years old, playing a musical instrument commonly used in classical music as a main subject; Pre-college students in years 3 or 4;

**Table 1. Distribution of the study centres and students participating in the study.**

| country | city | music university/conservatory | number of participants |
|---|---|---|---|
| Austria | Eisenstadt | Joseph Haydn Konservatorium | 6 |
| | Innsbruck | Tiroler Landeskonservatorium | 12 |
| | Linz | Anton Bruckner Privatuniversität | 9 |
| | Salzburg | Universität Mozarteum | 24 |
| | Wien | Universität für Musik und darstellende Kunst | 28 |
| Belgium | Antwerp | Royal Conservatoire Antwerp | 15 |
| | Hasselt | Robert Schumann Hochschule | 8 |
| | Namur | Institut supérieur de musique et de pédagogie | 14 |
| Denmark | Copenhagen | Royal Danish Academy of Music | 4 |
| | Odense | Danish National Music Academy | 4 |
| Estonia | Tallinn | Estonian Academy of Music and Theatre | 8 |
| Finland | Helsinki | Sibelius Academy (Uniarts) | 21 |
| France | Bordeaux | Pôle d'Enseignement Supérieur de la Musique et de la Danse | 8 |
| | Lille | École Supérieure Musique et Danse Hauts de France | 12 |
| | Paris | Conservatoire national supérieur de musique et de danse | 18 |
| Germany | Dresden | Hochschule für Musik Carl Maria von Weber | 19 |
| | Düsseldorf | Robert Schumann Hochschule | 17 |
| | Frankfurt | Hochschule für Musik und Darstellende Kunst | 30 |
| | Hamburg | Hochschule für Musik und Theater | 19 |
| | Karlsruhe | Hochschule für Musik Karlsruhe | 11 |
| | Leipzig | Hochschule für Musik und Theater Felix Mendelssohn Bartholdy | 32 |
| | Lübeck | Musikhochschule Lübeck | 19 |
| | Osnabrück | Universität Osnabrück | 29 |
| | Stuttgart | Hochschule für Musik und Darstellende Kunst | 12 |
| | Weimar | Hochschule für Musik Franz Liszt | 4 |
| Iceland | Reykjavík | Iceland University of Arts | 3 |
| Ireland | Cork | Cork School of Music | 40 |
| | Dublin | Royal Irish Academy of Music | 14 |
| Italy | Cast.Veneto | Conservatorio A. Steffani | 50 |
| | Ferrara | Conservatorio G. Frescobaldi | 55 |
| | Fiesole | Scuola di alto perfezionamento | 9 |
| | Milano | Conservatorio G. Verdi | 21 |
| | Novara | Conservatorio G. Cantelli | 24 |
| | Parma | Conservatorio A. Boito | 26 |
| | Piacenza | Conservatorio G. Nicolini | 35 |
| | Roma | Conservatorio S. Cecilia | 28 |
| | Salerno | Conservatorio G Martucci | 12 |
| Latvia | Riga | Jazeps Vitols Latvian Academy of Music | 14 |
| Portugal | Porto | Escola Superior de Música e Artes do Espetáculo | 5 |
| Scotland (UK) | Glasgow | Royal Conservatoire of Scotland | 11 |
| Spain | Alicante | Conservatorio Superior de Música Òscar Esplà | 10 |
| | Las Palmas | Conservatorio Superior de música de Canarias | 7 |
| | Madrid | Real Conservatorio Superior de Música | 22 |
| | Murcia | Conservatorio Superior de Música Manuel Massotti | 13 |
| | S. Sebastián | Musikene | 7 |
| | Sevilla | Conservatorio superior de música Manuel Castillo | 20 |
| | Vigo | Conservatorio Superior de Música | 9 |

**Table 1.** (Continued)

| country | city | music university/conservatory | number of participants |
|---|---|---|---|
| Sweden | Malmö | Malmö Academy of Music | 2 |
| | Göteborg | Academy of Music and Drama | 18 |
| | Stockholm | Royal College of Music—Kungliga Musikhögskolan | 12 |
| Switzerland | Basel | Hochschule für Musik | 9 |
| | Bern | Hochschule der Künste | 14 |
| | Lugano | Conservatorio della Svizzera italiana | 48 |
| | Luzern | Hochschule Luzern—Musik | 70 |
| The Netherlands | Amsterdam | Conservatorium van Amsterdam | 2 |
| | Maastricht | Conservatorium Maastricht | 4 |
| TOTAL | | | 997 |

Bachelor of Arts students in years 1, 2 and 3 and Master of Arts students in years 1, 2, 3, 4; students attending gap year programs or continuing education courses. Exclusion criteria were as follows: Composers and conductors; positive history of chronic and highly disabling neurological and/or rheumatic and/or psychological conditions in the last 12 months; surgery of the upper limbs and/or the spine in the last 12 months. All eligible students received an e-mail with information about the study, a participant information sheet with the electronic consent form and the link to the web-based questionnaire site. The student registries of the music universities and conservatories presented in Table 1 were used to distribute the aforementioned e-mail and thus to recruit the participants. A reminder e-mail was sent 3 weeks after the first e-mail.

## Outcome measures

This preliminary explorative study utilises a selection of the full menu of outcomes comprising the RISMUS project, which are available in the published protocol [47]. The use of questions and validated questionnaires allowed for a speculative exploration of suspected factors that were expected to be associated with a PRMD according to the current findings among the available literature.

The web-based survey included a bespoke questionnaire containing questions about: a) background and lifestyle (i.e. age, gender, self-reported height and weight, nationality, smoking status and sleeping habits); b) practice habits (i.e. main instrument, academic level, average time playing per week and years of experience, the perceived exertion after 45 minutes of practice without breaks [48], preparatory exercises and breaks during practice); c) health history (i.e. any painful MSK conditions, neurological and/or rheumatic and/or psychological disorders, surgery of the upper limbs and/or the spine or accidents/surgeries in the past 12 months and current medication) and the single question according to Zaza, Charles, and Muszynski [12] to identify the presence of PRMDs.

The *self-rated health (SRH) item* [49] was included for the assessment of health status, using a reliable and a valid [50] single-item measure ("In general, would you say your health is"), answered on a five-point scale from excellent to poor, with precedent amongst general population samples [51–53]. The short form of the *International Physical Activity Questionnaire (IPAQ-SF)* [54] was included for the assessment of physical activity participation levels. This widely used instrument for physical activity surveillance in adults (age range: 15–69 years old) [54–56] investigates the physical activity of four separate intensity levels (i.e. vigorous-intensity activity, moderate-intensity activity, walking, and sitting) with moderate to high relative

reliability (between 0.66 and 0.88). The *Kessler Psychological Distress Scale (K10)* [57] provides a reliable (kappa and weighted kappa scores range, 0.42 to 0.74) 10-item questionnaire of specific emotional states designed to measure anxiety and depression using five-level response scales (range: 10 to 50; 50 indicating the highest risk of anxiety or depressive disorder) [57]. Perfectionism among participants was assessed using the short form of the *Multidimensional Perfectionism Scale (HFMPS-SF)* [58–60], involving a 15-item questionnaire and rating for each on a 7-point Likert scale (from 1 "disagree" to 7 "agree"). Items are structured according to three subscales: self-oriented (SOP), other-oriented (OOP), and socially prescribed perfectionism (SPP), where higher scores on each scale, indicating higher levels of perfectionistic attitudes and behaviours (Cronbach α = 0.88, 0.74, and 0.81 for SOP, OOP, and SPP, respectively) [60]. Finally, the *Chalder Fatigue Scale (CFQ 11)* [61] was included for the assessment of fatigue and severity of tiredness. Each of eleven items are answered on a 4-point Likert-type scale (0 –asymptomatic- increasing to 3 as responses become more symptomatic), with higher global scores (range: 0 to 33) indicating greater tiredness and incorporating separate physical fatigue (items 1–7) and psychological fatigue [8–11].

According to their playing posture and arm position while playing, participants were allocated into six groups: music students playing musical instruments with both arms elevated in a frontal position (i.e. harp, trombone, and trumpet); music students playing musical instruments with both arms elevated in the left quadrant position (i.e. viola, violin); music students playing musical instruments with only the left arm elevated (i.e. cello, double bass); music students playing instruments with only the right arm elevated (i.e. flute, guitar); music students playing instruments in a neutral position, without the elevation of arms (i.e. accordion, bassoon, clarinet, euphonium/tuba; French horn, harpsichord, oboe, organ, percussion, piano, recorder, saxophone); singers. The arm position was classified as elevated when ≥40˚ abduction and/or ≥40˚ forward flexion occurred while playing. All other positions were categorised as neutral [21,62]. The current study used an original classification of risk associated with an elevated arm position (≥40˚) [62], but refined by the inclusion of two categories (i.e. "both arms elevated in a frontal position" and "both arms elevated in the left quadrant position") alongside "both arms elevated" [21]. Moreover, an additional category for singers has been employed due to the specific characteristics of their musical practice [63].

## Statistical analysis

Descriptive statistics were used to systematically summarise and present the data. For categorical variables, absolute and relative frequency distributions were presented. For continuous variables, since the normality test showed that all the variables considered were non-Gaussian, the median value and the range were used to summarise the variables.

Bivariate analysis was used to identify associations between the dependent variable MSK status and the covariates (i.e. demographic variables, as well as variables associated with health-related status and those associated with the playing of musical instruments) (see Table 2). According to their MSK status, participants were grouped into three sub-categories: (a) participants reporting no history of MSK conditions (*NoMSK*); (b) participants reporting MSK conditions related to musical practice (*PRMD*); [3] participants reporting MSK conditions not related to musical practice (*MSK*).

The distinction between the categories of MSK status was very important because it allowed descriptive contrast amongst factors associated with the general presence of MSK conditions (PRMDs or not) and factors specifically related to PRMDs. Since the MSK status variable was categorical, the statistical tests used were (a) chi-square test for verifying the associations with

**Table 2. Independent variables included in the study.**

| Type of variable | Name of variable |
|---|---|
| Demographic variables | Gender |
| | Age |
| | Nationality |
| | Academic level |
| Variables associated with health-related status | BMI |
| | Perceived health [SRH] |
| | Hours of sleep |
| | Smoking |
| | Medications |
| | Physical activity participation levels [IPAQ score] |
| | Psychological distress [K10 score] |
| | Perfectionism [HFMPS-SF: SO, OO, SP sub-scale score] |
| | Fatigue [CFQ 11 score] |
| Variables associated with the playing of musical instruments | Instrument [classification] |
| | Years of practice |
| | Hours of practice per day |
| | Perceived exertion after 45 minutes of practice without breaks |
| | Preparatory exercises |
| | Breaks during practice |

BMI, Body Mass Index; SRH, Self-rated health; IPAQ, International Physical Activity Questionnaire; K10, Kessler Psychological Distress Scale; HFMPS-SF, Multidimensional Perfectionism Scale–short form; SO, Self-oriented; OO, Other-oriented; SP, Socially prescribed; CFQ 11, Chalder Fatigue Scale.

categorical variables (b) Kruskal-Wallis tests for verifying the associations with continuous variables.

In addition, a multivariable analysis was conducted with an explorative aim in order to assess which candidate covariates were significantly associated with the three categories considered (i.e. *NoMSK; PRMD; MSK*) of the dependent variable MSK status. Since this variable was categorical, the multinomial logistic regression analysis was used.

Three models were explored for associated factors of PRMDs, with relative risk ratios (RRR), as the exponential of the multinomial logistic regression coefficient, used to indicate the relative probability for each candidate variable (RRR > 1 indicating that the greater probability of the outcome belonging within the comparison rather than reference group as the variable's scores increase, and vice versa). The models involved *PRMD*, *MSK* and *PRMD* as comparison groups, with *NoMSK*, *NoMSK* and *MSK* as corresponding reference groups.

Each model was estimated twice, using a stepwise approach with (a) forward selection: starting with an empty model (no variable included), the variables providing the most statistically significant improvement of the fit were progressively added until none of the remaining variables proved statistically significant (threshold for statistical significance: p-value below 5%); (b) backward elimination: starting with the full model (all variables included), the least significant variables were progressively eliminated until all the remaining variables were statistically significant (threshold for statistical significance: p-value below 5%).

The comparison of the estimates allowed the identification of four different kinds of factors: overall factors (i.e. variables statistically significant in all three models), MSK factors (i.e. variables statistically significant in the first two models but not in the third), PRMD factors (i.e.

variables statistically significant in the first and third models, but not in the second) and single factors (i.e. variables statistically significant in a single model only). Bivariate and multivariable analyses were performed on the overall sample and on the response of a sub-sample of participants not taking any supplements, contraceptives and/or actual medications to verify whether such an exogenous contribution could have biased the results or have influenced the responses.

## Results

Of the 997 participants agreeing to participate in the study by completing the informed consent, only 900 completed the whole web-based questionnaire. A total of 850 participants were included in the sample for the analysis (Fig 1).

A total of forty subjects were excluded from the analysis because they did not meet the inclusion criteria and 10 subjects were excluded because they were not able to determine if their MSK condition was a PRMD (i.e. interfered with their ability to play the instrument at the level to which they had been accustomed).

### Descriptive statistics

The following tables show descriptive features of the participants, including demographic variables (see Table 3), variables associated with self-reported health-related status (see Table 4) and variables associated with the playing of musical instruments (see Table 5).

Of the 850 participants, 11 played the accordion, 204 played a bowed instrument (violin, n = 117; viola, n = 24; cello, n = 44; double bass, n = 19), 90 a plucked instrument (guitar, n = 67; harp, n = 23), 142 a woodwind instrument (bassoon, n = 10; clarinet, n = 38; flute, n = 63; oboe, n = 21; recorder, n = 10), 101 a brass instrument (euphonium/tuba, n = 10; French horn, n = 20; saxophone, n = 26; trombone, n = 19; trumpet, n = 26), 28 percussion, 103 were singers, and 171 played the keyboards (harpsicord, n = 5; organ, n = 12; piano, n = 154).

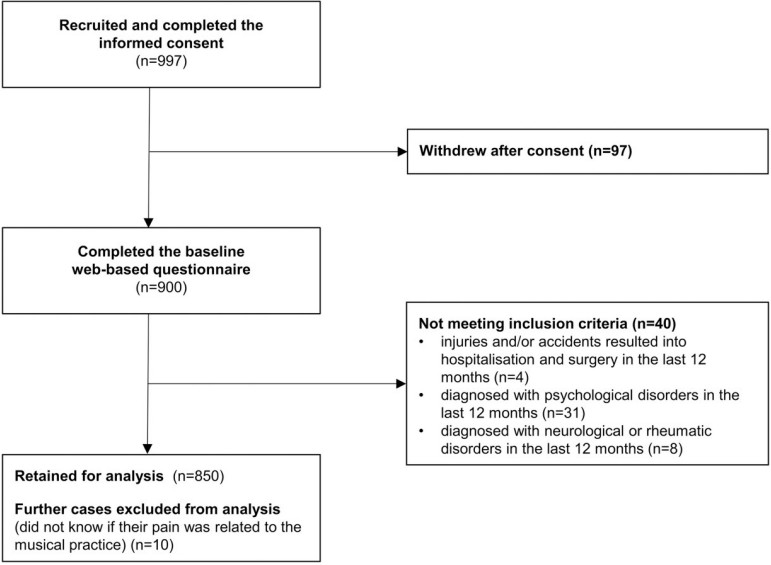

**Fig 1. Flowchart of participant selection for the analysis.**

**Table 3. Descriptive statistics of demographic variables.**

| Variable | | n | % |
|---|---|---|---|
| **Gender** | Female | 522 | 61.4% |
| (n = 850) | Male | 325 | 38.3% |
| | Other | 3 | 0.4% |
| **Age** | median | 22 | |
| (n = 850) | range | 18–48 | |
| **Nationality (region)**[*] | South Europe | 386 | 45.4% |
| (n = 850) | West Europe | 312 | 36.7% |
| | North Europe | 81 | 9.5% |
| | East Europe | 35 | 4.1% |
| | Other | 36 | 4.2% |
| **Academic level** | Pre-college | 86 | 10.1% |
| (n = 850) | Bachelors 1&2 | 150 | 17.6% |
| | Bachelors 3&4 | 171 | 20.1% |
| | Masters 1&2 | 124 | 14.6% |
| | Masters 3&4 | 174 | 20.5% |
| | Gap year/continuing education | 145 | 17.1% |

[*]This classification was made according to United Nations, S. D. *Standard Country or Area Codes for Statistical Use*, *Series M*, *No. 49 (M49)* <https://unstats.un.org/unsd/methodology/m49/> (1999).

Table 6 includes the distribution of participants, according to the six-year group levels and the six types of instruments' classification.

In total, the prevalence of participants with a self-reported PRMD was 48% (n = 408), while less than 20% self-reported a MSK condition that was not related to the musical practice, and about one third self-reported having no MSK condition (Fig 2).

Participants playing musical instruments with both arms elevated in a frontal position self-reported the highest prevalence of PRMDs (54.4%), followed by participants playing instruments with only the right arm elevated (51.1%) and with both arms elevated in the left quadrant position (50.4%). Participants playing instruments in a neutral position (i.e. without the elevation of the arms) self-reported a prevalence of 47.7% of PRMDs. Participants playing musical instruments with only the left arm elevated and singers self-reported a similar prevalence of PRMDs, almost 43% and 41% respectively (Fig 3).

## Bivariate and multivariable analyses

Results of bivariate and explorative multivariable analyses derived from the overall sample and from a sub-sample of participants not taking any supplements, contraceptives and/or actual medications, did not reveal any significant variations or differences. This similarity amongst the findings indicated that the latter factors had not intruded substantively and accordingly, the overall sample's results have been reported for simplicity. Statistically significant relations with the MSK status variable emerged for eight of the 21 variables considered (see Table 7).

Nationality, academic level, perfectionism, fatigue, years of practice and perceived exertion after 45 minutes of practice without breaks ($\chi^2_{(df,\ 2\ to\ 10)}$ = 10.4 to 49.5; p<0.001), as well as psychological distress ($\chi^2_{(df,\ 2)}$ = 8.4; p<0.01) were related significantly with MSK status (*NoMSK*, *PRMD* and *MSK*). Participants from countries in West Europe self-reported the second-highest prevalence of PRMDs (52%) but simultaneously the lowest prevalence of MSK conditions that did not interfere with their playing ability (8%) (see Table 7). By contrast,

**Table 4. Descriptive statistics of variables associated with self-reported health-related status.**

| Variable | | n | % |
|---|---|---|---|
| **BMI in kg/m$^2$** | median | 21.5 | |
| (n = 828) | range | 15.3–41.0 | |
| **Perceived health [SRH]** | Excellent | 65 | 7.6% |
| (n = 850) | Very good | 266 | 31.3% |
| | Good | 389 | 45.8% |
| | Fair | 117 | 13.8% |
| | Poor | 13 | 1.5% |
| **Hours of sleep** | median | 7 | |
| (n = 849) | range | 4–10 | |
| **Smoking** | Yes | 131 | 15.5% |
| (n = 848) | No | 717 | 84.5% |
| **Medications** | Nothing | 710 | 83.5% |
| (n = 850) | Supplement/contraceptive | 60 | 7.1% |
| | Medicine | 80 | 9.4% |
| **Physical activity participation levels [IPAQ score]** | High | 153 | 18.2% |
| (n = 843) | Moderate | 415 | 49.2% |
| | Low | 275 | 32.6% |
| **Psychological distress [K10 score]** | median | 20.0 | |
| (n = 843) | range | 10–46 | |
| **Perfectionism [HFMPS-SF score]** | | | |
| SO sub-scale score | median | 25.0 | |
| (n = 830) | range | 5–35 | |
| OO sub-scale score | median | 18.0 | |
| (n = 838) | range | 5–35 | |
| SP sub-scale score | median | 17.0 | |
| (n = 836) | range | 5–35 | |
| **Fatigue [CFQ 11 score]** | median | 13.0 | |
| (n = 825) | range | 0–33 | |

BMI, Body Mass Index; SRH, Self-rated health; IPAQ, International Physical Activity Questionnaire; K10, Kessler Psychological Distress Scale; HFMPS-SF, Multidimensional Perfectionism Scale–short form; SO, Self-oriented; OO, Other-oriented; SP, Socially prescribed; CFQ 11, Chalder Fatigue Scale.

participants from East Europe self-reported the highest prevalence of PRMDs (54%), a higher level of MSK conditions that did not interfere with their playing ability (31%), but also the lowest level of no MSK conditions (16%). Furthermore, students at the Pre-college academic level self-reported the highest prevalence of no MSK conditions (45%), while first- or second-year Masters students were notable for having the highest level of PRMDs (64%). Similarly, participants reporting the highest number of years of practice (14 years), highest perceived exertion after 45 minutes of practice without breaks (5 units), as well as the highest fatigue level (14 units) were also associated with reporting the prevalence of PRMDs. In general, the highest scores recorded for psychological distress [21] and perfectionism [18] were associated with participants reporting a MSK condition (including PRMDs).

Table 8 reports the RRR for each variable included within the models of the multivariable analysis. The pseudo-R$^2$ (Cox-Snell, Cragg-Uhler/Nagelkerke) ranged from 0.11 to 0.19, indicating moderate accuracy amongst the models. An acceptable goodness-of-fit (0.70 to 0.80) [64] was confirmed by the separate logistic regression estimates of the three models, for which

**Table 5. Descriptive statistics of variables associated with the playing of musical instruments.**

| Variable | | n | % |
|---|---|---|---|
| **Instrument** | Elevated both frontal | 68 | 8.0% |
| [classification] | Elevated both left | 141 | 16.6% |
| (n = 850) | Elevated left | 63 | 7.4% |
| | Elevated right | 131 | 15.4% |
| | Neutral | 344 | 40.5% |
| | Singers | 103 | 12.1% |
| **Years of practice** | median | 13 | |
| (n = 850) | range | 6–35 | |
| **Hours of practice per day** | median | 3 | |
| (n = 849) | range | 3–8 | |
| **Perceived exertion after 45 minutes of practice without breaks** | Median range | 4 0–10 | |
| (n = 843) | | | |
| **Preparatory exercises** | Yes | 354 | 41.7% |
| (n = 850) | No | 496 | 58.3% |
| **Breaks during practice** | Yes | 522 | 61.4% |
| (n = 850) | No | 328 | 38.6% |

Elevated both frontal: Music students playing musical instruments with both arms elevated in a frontal position (i.e. harp, trombone, and trumpet); Elevated both left: Music students playing musical instruments with both arms elevated in the left quadrant position (i.e. viola, violin); Elevated left: Music students playing musical instruments with only the left arm elevated (i.e. cello, double bass); Elevated right: Music students playing instruments with only the right arm elevated (i.e. flute, guitar); Neutral: Music students playing instruments in a neutral position, without the elevation of arms (i.e. accordion, bassoon, clarinet, euphonium/tuba; French horn, harpsichord, oboe, organ, percussion, piano, recorder, saxophone).

the area under the ROC curve ranged from 0.70 to 0.75. In addition, no multicollinearity has been identified (average variance inflation factor between 1.02 and 1.06, depending on the model).

The analysis identified four different kinds of factors. The variable Nationality West Europe was the only overall factor that appeared statistically significant in all three models. For instance, as can be seen in Table 8 in the first model *PRMD vs NoMSK* (i.e. first column, where *PRMD* is the comparison group and *MSK* is the reference group), the RRR for West Europe equals 0.647, meaning that the probability of belonging within the comparison group is about 35% [This percentage was calculated according to the following formula: (0.647–1) · 100 = -35.3%] lower for Western European participants compared to Southern European participants, keeping all the other variables constant. By contrast, the direction changed in the focal model (i.e. third column *PRMD vs MSK*), showing that Western European participants had a higher probability (RRR = 4.524; RRR > 1) of belonging within the comparison group. On the other hand, the MSK factors (i.e. variables statistically significant in the first two models but not in the third) were found to be perceived health [SRH] (RRR = 1.104; RRR>1) and fatigue [CFQ 11 score] (RRR = 1.084) and thus related to the presence of a MSK condition in general but not specifically to the presence of PRMD. Moreover, PRMD factors (i.e. variables statistically significant in the first and third models, but not in the second) were found to be years of practice (RRR = 1.040; RRR>1) and perceived exertion after 45 minutes of practice without breaks (RRR = 1.044; RRR>1), suggesting that these factors were related to the specific presence of PRMD. Finally, there was only one PRMD-related single factor and was the variable academic level Masters 1&2 (RRR = 2.747; RRR>1), which appeared statistically significant in

**Table 6. The distribution of participants, according to the six-year group levels and the classification of instruments.**

| | n participants | | | | | |
| --- | --- | --- | --- | --- | --- | --- |
| *category* | Pre-college | 1&2 BA | 3&4 BA | 1&2 MA | 3&4 MA | Gap year/cont. education |
| Both arms elevated frontal (n = 68) | 17 | 13 | 11 | 14 | 6 | 7 |
| Both arms elevated left (n = 141) | 12 | 24 | 22 | 23 | 35 | 25 |
| Left arm elevated (n = 63) | 7 | 12 | 12 | 9 | 11 | 12 |
| Right arm elevated (n = 131) | 9 | 25 | 28 | 17 | 32 | 20 |
| Neutral (n = 344) | 32 | 58 | 74 | 40 | 74 | 66 |
| Singers (n = 103) | 9 | 18 | 24 | 21 | 16 | 15 |
| TOTAL | 86 | 150 | 171 | 124 | 174 | 145 |

1&2 BA: Music students enrolled in their first and second year of Bachelor of Arts in Music; 3&4 BA: Music students enrolled in their third and fourth year of Bachelor of Arts in Music; 1&2 MA: Music students enrolled in their first and second year of Master of Arts in Music; 3&4 MA: Music students enrolled in their third and fourth year of Master of Arts in Music; Gap year/cont.education: Music students experiencing a gap year or enrolled in a continuing education programme.

the first model *PRMD vs NoMSK*. When compared to Pre-college, students attending the 1st and 2nd year of Masters had a higher probability of belonging within the comparison group (i.e. *PRMD*) compared to not having any MSK conditions.

## Discussion

This study focused on the prevalence of PRMDs in a large-scale study population of music students at different educational stages (i.e. university-level students and Pre-college students)

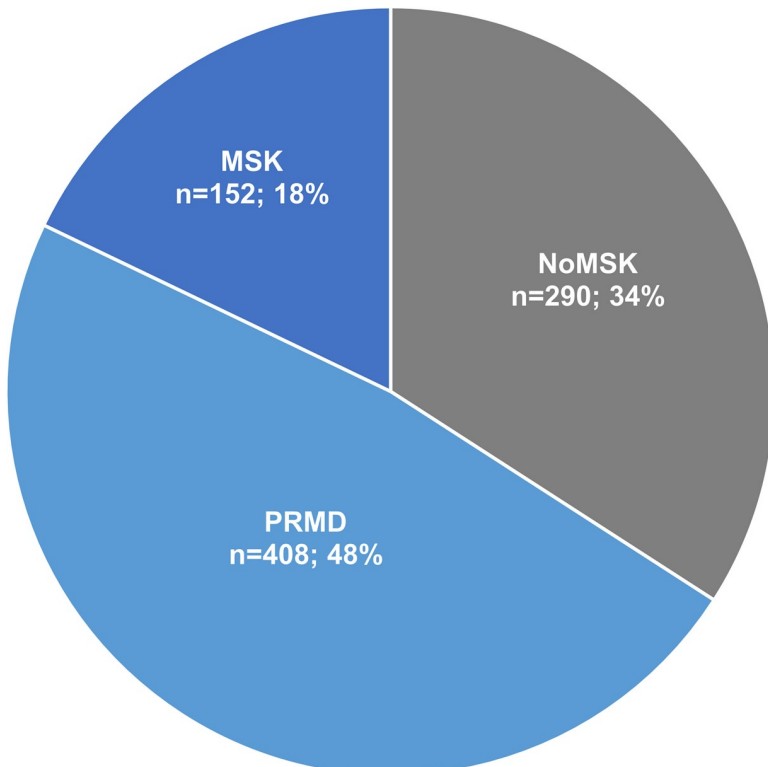

**Fig 2. Musculoskeletal status among participants.** Prevalence of self-reported playing-related musculoskeletal disorders (PRMDs, n = 408; 48%), self-reported musculoskeletal condition not related to the musical practice (MSK, n = 152; 18%) and musculoskeletal condition (MSK, n = 152; 18%). PRMDs, Playing-related Musculoskeletal Disorders; MSK, Musculoskeletal.

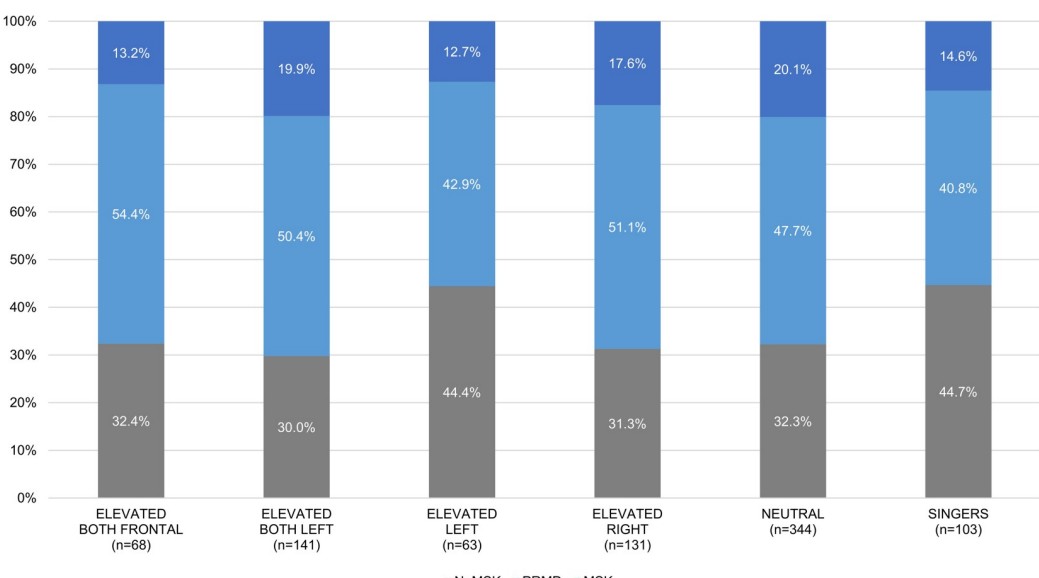

**Fig 3. Prevalence of self-reported playing-related musculoskeletal disorders (PRMDs) among groups according to their playing posture and arm position while playing.**

and enrolled in different pan-European music institutions at baseline of the RISMUS project. Music students participating in this novel large-scale study involving 20 European countries self-reported a high prevalence of painful MSK conditions (65%), of those 48% self-reported PRMDs.

A further goal was to begin to identify variables that might be associated with the self-reported presence of PRMDs among this population that ultimately would facilitate future longitudinal analyses. Results highlighted that coming from West Europe, being a first- or a second-year Masters student, having more years of experience and higher rates of perceived exertion after 45 minutes of practice without breaks were factors significantly associated with self-reported presence of PRMD. In this regard, the current study integrates novel and robust descriptive data with explorative and speculative analyses via relatively sophisticated statistical modelling for factors that may be associated with PRMDs (i.e. multinomial regression model).

The present study's findings can be contextualised with comparison to those from Pre-college participants, who offer a controlled reference as students who have not yet been clearly orientated towards a musical career by means of a university education. It could be argued that this group of participants were compromised as experimental controls reflecting the responses of the general public, as they inevitably undertake preparatory training in musicianship [65]. Nevertheless, they would not yet have undergone the requisite higher demands and more intense training to further work in the highly competitive musical profession. As such, Pre-college participants offered a reasonable compromise in regard to the likely responses of the general population, while simultaneously allowing this study to remain congruent with both Zaza et al.'s restrictive definition of only musicians being eligible to be afflicted by PRMDs, and a distinction between PRMDs and non-PRMDs in accordance with recommendations from the performing arts medicine field [18,30,31]. Indeed, nearly half of Pre-college participants self-reported having no MSK conditions (45%), as can be seen in Table 7. One of the most prominent findings indicated that between the different academic levels, the prevalence of PRMD had peaked within the Masters 1&2-year group (64%) having been recorded at more modest levels within the Pre-college group (44%) and the Masters 3&4-year group (43%). Students

**Table 7. Bivariate associations between MSK status and factors reflecting demographics, health-related status and the playing of musical instrument.**

| | MSK status | | | Statistical test result |
|---|---|---|---|---|
| | **NoMSK** | **PRMD** | **MSK** | |
| **Nationality (region)** | | | | |
| South Europe | 30% | 45% | 25% | $\chi^2$ (df, 8) = 46.8*** |
| West Europe | 40% | 52% | 8% | |
| North Europe | 37% | 47% | 16% | |
| East Europe | 14% | 54% | 31% | |
| Other | 42% | 42% | 16% | |
| *Total* | 34% | 48% | 18% | |
| **Academic level** | | | | |
| Pre-college | 45% | 44% | 11% | $\chi^2$ (df, 10) = 28.0*** |
| Bachelors 1&2 | 36% | 50% | 14% | |
| Bachelors 3&4 | 34% | 44% | 22% | |
| Masters 1&2 | 25% | 64% | 11% | |
| Masters 3&4 | 36% | 43% | 21% | |
| Gap year/continuing education | 31% | 46% | 23% | |
| *Total* | 34% | 48% | 18% | |
| **Psychological distress** | | | | |
| [K10 score] | | | | |
| Median (range) | 19 (10–46) | 20 (10–45) | 21 (10–44) | $\chi^2$ (df, 2) = 8.4** |
| **Perfectionism [HFMPS-SF]** | | | | |
| SP sub-scale score | | | | |
| Median (range) | 16 (5–33) | 18 (5–35) | 18 (5–35) | $\chi^2$ (df, 2) = 12.4*** |
| **Fatigue [CFQ 11 score]** | | | | |
| Median (range) | 11 (1–28) | 14 (0–33) | 13 (2–33) | $\chi^2$ (df, 2) = 49.5*** |
| **Years of practice** | | | | |
| Median (range) | 12 (6–35) | 14 (6–34) | 13 (6–28) | $\chi^2$ (df, 2) = 10.4*** |
| **Perceived exertion after 45 minutes of practice without breaks** | | | | |
| Median (range) | 4 (0–10) | 5 (0–10) | 4 (0–10) | $\chi^2$ (df, 2) = 18.9*** |

*** $p < 0.001$

** $p < 0.01$

* $p < 0.05$.

For categorical variables, the musculoskeletal (MSK) status relative distributions (row percentages) for every category of the variable considered has been reported, as well as the chi-square statistic and its statistical significance level. For continuous variables, the median and the range for each MSK status category has been reported, as well as the chi-square statistic of the Kruskal-Wallis test and its statistical significance level.

MSK, Musculoskeletal; SRH, Self-rated health; K10, Kessler Psychological Distress Scale; HFMPS-SF, Multidimensional Perfectionism Scale–short form; SP, Socially prescribed; CFQ 11, Chalder Fatigue Scale.

undertaking subsequent gap years or further study recorded an intermediate level of PRMD prevalence (46%). Future RISMUS analyses will corroborate the longitudinal patterning of these findings. Nevertheless, the present results are consistent with a recent study that reported a prevalence of playing-related health problems varying between 29% at the beginning of their university training and 42% among second year students, that later decreased to 36% in their third year [66].

Remarkably, the peak in prevalence of PRMDs amongst first or second year Masters students when collated with their non-PRMDs, contributed a prevalence of MSK conditions of 75%. The latter group's prominence in this regard was also confirmed by the multinomial

**Table 8. Multinomial logistic regression analysis of associations between MSK status and factors reflecting demographics, health-related status and the playing of musical instrument.**

| Variables | PRMD vs NoMSK | MSK vs NoMSK | PRMD vs MSK |
|---|---|---|---|
| **Nationality** (reference category: South Europe) | | | |
| West Europe | 0.647* | 0.220*** | 4.524*** |
| | (0.125) | (0.061) | (1.196) |
| North Europe | 0.589 | 0.410* | 1.882 |
| | (0.180) | (0.157) | (0.684) |
| East Europe | 2.133 | 2.344 | 0.391 |
| | (1.140) | (1.352) | (0.219) |
| Other | 0.615 | 0.456 | 2.167 |
| | (0.254) | (0.235) | (1.089) |
| **Academic level** (reference category: Pre college) | | | |
| Bachelors 1&2 | 1.504 | 1.776 | - |
| | (0.460) | (0.833) | - |
| Bachelors 3&4 | 1.271 | 2.210 | - |
| | (0.388) | (0.987) | - |
| Masters 1&2 | 2.747** | 2.408 | - |
| | (0.938) | (1.252) | - |
| Masters 3&4 | 1.079 | 1.875 | - |
| | (0.337) | (0.837) | - |
| Gap year/ | 1.302 | 2.811* | - |
| continuing education | (0.428) | (1.286) | - |
| **Perceived health [SRH]** (reference category: Excellent) | | | |
| Very good | 1.387 | 2.547 | - |
| | (0.445) | (1.727) | - |
| Good | 1.766 | 3.188* | - |
| | (0.549) | (1.560) | - |
| Fair or poor | 2.166* | 3.799* | - |
| | (0.792) | (2.067) | - |
| **Perfectionism [HFMPS-SF]** | | | |
| **OO sub-scale score** | - | 1.041* | - |
| | - | (0.019) | - |
| **Fatigue [CFQ11 score]** | 1.104*** | 1.084*** | - |
| | (0.019) | (0.023) | - |
| **Years of practice** | 1.040* | - | 1.044* |
| | (0.020) | - | (0.022) |
| **Perceived exertion after 45 minutes of practice without breaks** | 1.009* | - | 1.011* |
| | (0.004) | - | (0.004) |
| Constant | 0.085*** | 0.026*** | 0.621 |
| | (0.043) | (0.019) | (0.216) |

*** p<0.001

** p<0.01

* p<0.05.

The values reported in the table are the relative risk ratios (RRR) and the standard errors, which are indicated in parentheses. The RRR indicates how the probability of belonging within the comparison group (the first in the column) relative to the probability of belonging within the reference group changes with the variable considered. In the first column, the comparison is *PRMD* and the reference is *NoMSK*. In the second column, the comparison is *MSK* and the reference is *NoMSK*. In the third column, the comparison is *PRMD* and the reference is *MSK*. An RRR > 1 indicates that the probability of belonging within the comparison group relative to the probability of belonging within the reference group increases as the value of the variable increases, while it is the opposite for an RRR < 1.

PRMDs, Playing-related Musculoskeletal Disorders; MSK, Musculoskeletal; SRH, Self-rated health; K10, Kessler Psychological Distress Scale; HFMPS-SF, Multidimensional Perfectionism Scale–short form; OO, Other Oriented; CFQ 11, Chalder Fatigue Scale.

logistic regression analysis in which, when compared to Pre-college, students attending the 1st and 2nd year of a Masters course were associated with having a higher self-reported prevalence of PRMD (RRR > 1). This trend may be attributed to the fact that the transition to higher musical training (i.e. Masters studies) often requires an increase of practising' hours to deal with higher demands, such as the ability to compete with others [66], tolerance and perseverance and the ability to develop an effective strategy for self-assessment. These are indispensable attributes for any aspiring musician in order to pass the difficult entrance examination, and to become familiarised with the higher performance demands that will be inevitable.

It was also notable that a peak in prevalence was recorded by students at the early stages of their Masters level education (Masters 1&2), and not amongst students at Masters 3&4. It would be interesting to speculate that progression to a third year of a Masters level education might represent a critical juncture at which students become either increasingly accustomed to the high levels and intensities of practice in order to reduce their risk of acquiring a PRMD, or similarly, change their playing technique to accommodate the effects of past MSK conditions. In addition, another possible reason for the reduction of PRMDs' prevalence among Masters students at later stages could be that, although the literature reports that musicians engage poorly in health promoting behaviours [43,67–69], courses and short-term health education programs have been recently developed to integrate useful insight from health professionals as well as knowledge from relevant health education settings [40,42,45]. Students at later stages could have had the possibility to engage in these useful programs and reduce or treat their painful condition. In addition, understanding potential mechanisms underpinning elevated prevalence of PRMD may be critical because approximately 12% of musicians abandon their musical careers due to such problems [17,70].

The patterns of prevalence for PRMDs during musicians' education may also be related to different aspects of fatigue and physical exertion. In our findings, the median of CFQ 11 for the physical and psychological fatigue assessment [71] and the median of the perceived exertion after 45 minutes of practice without breaks were significantly higher among participants reporting PRMDs, suggesting that there was a possible relationship between these variables and playing-related conditions. In fact, if we consider *PRMD vs NoMSK* (comparison group: *PRMD*; reference group: *NoMSK*) in Table 8, it can be seen that CFQ 11 score was a statistically significant factor, and thus the probability of having a PRMD compared to not having any MSK condition increases by a factor of 1.104 (approximately 10%) for each additional point of the CFQ 11 score, keeping all the other variables constant. Nevertheless, these findings should be considered cautiously as they reflect speculative logistic regression modelling of multiple candidate variables within a cross-sectional design involving necessarily self-reported data.

Previous research regarding the effect of pain on muscle fatigue has reported that pain significantly influences fatigue [72–74]. Another research study has shown that accomplishing peak performance depends on effective fatigue' management, taking into account both fatigue and recovery processes [75]. In addition, despite the similarity of physical demands between musicians and athletes, in sport, periodisation is used to adapt the intensity, length and frequency of physical loading to optimise continuous development of performance, without excessive exertion that may increase the risk of injury for athletes [76]. Unfortunately, such approaches based on periodisation are not familiar concepts in musical settings, where rehearsal and performance schedules for instrumentalists are typically organised without any concern for physical loading and the guidelines for fatigue management are generally ignored in the musical environment. For instance, according to Rickert et al. [77], musicians often have a low-level of "control" over intensity of practice time, repertoire and busy schedules that may in turn lead to increased stress and physical effort. In fact, as can be seen in Table 8, the perceived exertion after 45 minutes practice without breaks (RRR > 1) was statistically

significant in the *PRMD vs NoMSK* and *PRMD vs MSK* comparisons, but not in the *MSK vs NoMSK* comparison, suggesting that this factor might be related to the specific presence of PRMD, although a further longitudinal analysis will allow a careful evaluation of this important aspect.

In regard to a wider perspective on health-related artistic accomplishment and the impact of injury on participation, our findings have shown that, when compared to the reference category of having "excellent" health, the category "fair or poor" was associated with having a higher self-reported prevalence of a MSK condition (PRMD or not) (RRR > 1; see Table 8). These findings indicate that the impact of PRMDs on students' health may be highly significant and are in line with previous evidence that painful MSK conditions may be related to a lower perception of life-quality and hamper playing-quality [31]. For instance, a similar picture is provided by other studies that have investigated health perception among music students, who rated their health worse compared to an age-and sex matched group of students who did not play music and reported worst behaviour records of health responsibility [37,43,67,69]. Similarly, Rickert et al. [78] reported an insufficient health awareness of injury among students playing the cello and Kreutz et al. [69] showed poor stress management, inadequate nutrition and low levels of health responsibility among music students, suggesting a consistent need for continuing to develop strategies to enhance health support as an essential aspect of conservatoire and music university education by for instance integrating it into students' curricula and learning programs [43,78,79]. During their professional training, music students should learn how to cope with physical and psychological demands with the help of preventive measures. Body-oriented courses (i.e. posture, strength and conditioning exercises) and relaxation techniques, as well as psychological programs for stress and wellbeing have been shown to have a preventive effect [42,44,45,80]. This indicates that better results on MSK conditions among music students could be obtained by addressing health awareness and attitudes to injury at the university or even at the Pre-college. Indeed, music universities represent the primary channel for the improvement of health awareness and the implementation of injury prevention initiatives, being an important gateway to the professional world [81]. Therefore, strengthening attitudes and behaviours toward health music making will create a step change in educational and employment contexts, shaping future practice and addressing injury prevention to possibly avoid or at least reduce incidences of PRMDs. According to Rickert et al. [78] and Spahn et al. [80], health behaviours toward prevention may be easier to be addressed in the younger generation of musicians who may not already have such established habits. Preventive courses and health promotion among musicians should start already at the beginning of their musical training, with the objective to protect music students from PRMDs during their studies and to prepare them for the future professional demands. For instance, music students without a disorder at the beginning of their professional education would benefit of an increasing sensitisation in health promotion and injury prevention. On the other hand, students already suffering from health concerns need to be informed about potential strategies to reduce symptoms [80].

Consistent with previous studies [30,63], there was no statistical evidence of an association between PRMDs and instruments' classification. Despite the large size of our study's sample, instrument-specific analyses were not viable statistically, and anatomically-relevant categories of playing position were used instead [21,62]. Participants playing musical instruments with both arms elevated in a frontal position self-reported the highest prevalence of PRMDs (54.4%) and singers self-reported the lowest prevalence (40.8%). In the previous literature, playing string versus other instruments [12,23,29,82] and with elevated arms [21,62] provoked higher prevalence. It is plausible that any conflict amongst these findings may be attributed to heterogeneity of instrument group' classification or restricted study sample sizes with the contemporary literature. As such, evidence from future studies involving large, instrument-

specific populations or consensus classification would facilitate meta-analytical synthesis and further understanding of the effects of biomechanical stress [31].

The regional distribution of the prevalence of PRMDs appears to be relatively homogeneous, despite East and West European participants self-reporting slightly higher rates (54% and 52%, respectively). In addition, West Europeans also self-reported lower prevalence of non-playing related disorders (8%) compared with East European counterparts (31%). This finding was corroborated by multinomial regression analyses, in which Western European participants had a lower probability of having a MSK condition (RRR<1) compared to Eastern Europeans, but a higher probability of having a PRMD relative to having a generic MSK condition (RRR>1; model 2) with relatively greater perceived interference with musical performance. It may be speculated that West European participants tended to suffer less from MSK conditions than their East European counterparts due to preventative interventions being more common in this region [11,37,66,83]. Future studies might explore music students' health education and health-related behaviours in order to further understand their potential impact on PRMD prevalence and impact. For instance, it is plausible that participants' origins might be considered as an important factor because knowing where participants have lived most of their lives can provide important information about their experience with regard to their instrumental practice and cultural preferences, and thus assessing the probability for developing a PRMD. These results could be employed to develop or improve targeted initiatives for prevention to improve musical performance and to enhance physical endurance, while avoiding overuse injuries and reducing muscular fatigue.

## Limitations

There are limitations to be aware of when considering the findings. Firstly, the study used self-reported data without any physical examination to formally exclude any serious diseases that affect the musculoskeletal system. Nonetheless, the self-reported data was used in the best way possible to exclude some participants who had reported either histories of neurological, rheumatic and psychological disorders, or recent surgeries to the upper limb or spine, in order to ensure that the sample comprised only "healthy" participants. In addition, bivariate and multivariable analyses were performed on the overall sample and on a sub-sample of participants not taking any supplements, contraceptives and/or actual medications to verify whether such an exogenous contribution could have biased the results or have influenced the responses.

Secondly, this study used a web-based questionnaire that has the benefit of being able to reach the widest range of potential respondents in a more cost effective and safe way, but this could also represent a limitation. Furthermore, the invitation for participants to complete the questionnaire was sent by the school registries and not by the researchers, without the possibility of reinforcing the invitation by sending a reminder in another form (e.g. via a telephone interview). In addition, relevant information from non-respondents had not been accessible, which could have been used to assess for the intrusion of biases within the study's results. However, the sample size was quite large and this could be considered as adding robustness to the study's findings and enhancing the facilitating knowledge about the prevalence and development of PRMDs.

Furthermore, this study was performed amongst music students without a control group of non-musicians. However, as the distinction between PRMDs and non-PRMDs had been purposely emphasised within this study, this aspect could not have been achieved by including and considering the responses from a group of non-musicians. As described previously [47], Pre-college students, who would have been expected to have the least experience of musicianship, acted as a reference group.

Moreover, another limitation consists of the impossibility to control information on the individual and/or the institutional level of behaviours or attitudes toward prevention. For instance, engaging in health-prevention programs could represent a potential confounder that might have affected our results. However, the web-based questionnaire includes questions on strategies to reduce any MSK conditions they may have had in the past and thanks to the replies of the two follow-ups we will have more information and we will be able to record this important aspect. In addition, the participatory level of physical activity has been monitored with the International Physical Activity Questionnaire (IPAQ), which is a well-known measure to offer data on health–related physical activity. Nonetheless, whereas it is important to consider individual health-promoting behaviours [68,69], Perkins and colleagues [43] suggested that there is still the need to continue evaluating health behaviours and awareness among students and teachers inside music institutions, as well as environmental factors that might be perceived hampering or facilitating health and prevention. It is plausible to think that the environmental factors might be to some extent changed to accommodate research findings regarding the prevention of MSK conditions.

Furthermore, the authors cannot exclude a potential sampling bias as the information concerning the number of students enrolled in each school participating in the study is not available because it consists of confidential data, without a formal permission to publish.

Finally, the present explorative research study did not encompass complete coverage of all the potential factors contributing to precision within multinomial regression analyses predicting PRMDs in music students. Nonetheless, the models offered acceptable statistical power, absence of any multicollinearity and acceptable goodness of fit (0.70 to 0.80) [64]. The latter metric in particular suggests that other factors that were outside of the scope of this study, were influencing prevalence of PRMDs, and should be considered within future research. In summary, although the results of this study were exploratory, a large and varied sample of music students from different parts of Europe has been examined, constituting one of the largest studies in the performing arts medicine. In addition, a relatively sophisticated statistical modelling with an explorative perspective to identify factors that may be associated with PRMDs has been used. Examining the baseline data is an initial and necessary exploratory step toward better characterising the study population and the characteristics associated with self-reported PRMDs. It will help to guide further examination of our sample from a longitudinal perspective to determine the relative stability of these initial findings over time.

## Conclusions

The high prevalence of PRMDs among music students, especially those studying at university-level, has been confirmed in this study and associated factors have been identified, highlighting the need for relevant targeted interventions as well as effective prevention and treatment strategies.

Although the results of this study should be interpreted with caution due to the cross-sectional and self-reported nature of the data, they reflect the findings from a relatively large-scale investigation involving multiple centres across Europe and importantly, students at different stages of their education (from Pre-college to Masters levels). These findings may contribute important adjunct findings to those from the antecedent literature facilitating effective approaches towards primary prevention of PRMDs and their associated burden among music students and professionals. They may usefully raise awareness further within the musical and scientific communities.

## Acknowledgments

We wish to thank the participating music students and the study centres in helping recruiting the participants, as well as Alessandro Chiarotto for his assistance in selecting the assessment measures and Alessandro Schneebeli for his assistance in the classification of instruments according to their position. In addition, we would like to thank Andrea Cavicchioli and Paola Di Giulio for their assistance in the classification of the medicines during the analysis of data.

## Author Contributions

**Conceptualization:** Cinzia Cruder, Marco Barbero, Pelagia Koufaki, Nigel Gleeson.

**Data curation:** Cinzia Cruder, Marco Barbero.

**Formal analysis:** Emiliano Soldini.

**Funding acquisition:** Cinzia Cruder, Marco Barbero.

**Investigation:** Cinzia Cruder.

**Methodology:** Cinzia Cruder, Marco Barbero, Pelagia Koufaki, Emiliano Soldini, Nigel Gleeson.

**Project administration:** Cinzia Cruder, Marco Barbero.

**Resources:** Cinzia Cruder.

**Supervision:** Marco Barbero, Nigel Gleeson.

**Validation:** Nigel Gleeson.

**Writing – original draft:** Cinzia Cruder.

**Writing – review & editing:** Cinzia Cruder, Pelagia Koufaki, Emiliano Soldini, Nigel Gleeson.

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
