## [Decision Letter · Decision Letter 0]

16 Sep 2020

PONE-D-20-17856

Prevalence and associated factors of playing-related musculoskeletal disorders among European music students. Baseline findings from the Risk of Music Students (RISMUS) longitudinal study

PLOS ONE

Dear Dr. Cruder,

Thank you for submitting your manuscript to PLOS ONE. After careful consideration, we feel that it has merit but does not fully meet PLOS ONE’s publication criteria as it currently stands. Therefore, we invite you to submit a revised version of the manuscript that addresses the points raised during the review process.

We look forward to receiving your revised manuscript.

Kind regards,

Feng Pan, M.D

Academic Editor

PLOS ONE

2. Please include additional information regarding the survey or questionnaire used in the study and ensure that you have provided sufficient details that others could replicate the analyses.

For instance, if you developed a questionnaire as part of this study and it is not under a copyright more restrictive than CC-BY, please include a copy, in both the original language and English, as Supporting Information. Moreover, please include more details on how the questionnaire was pre-tested, and whether it was validated.

Reviewers' comments:

Reviewer's Responses to Questions

**Comments to the Author**

1. Is the manuscript technically sound, and do the data support the conclusions?

Reviewer #1: Yes

Reviewer #2: Yes

Reviewer #3: Partly

2. Has the statistical analysis been performed appropriately and rigorously? 

Reviewer #1: Yes

Reviewer #2: Yes

Reviewer #3: I Don't Know

3. Have the authors made all data underlying the findings in their manuscript fully available?

Reviewer #1: No

Reviewer #2: Yes

Reviewer #3: Yes

4. Is the manuscript presented in an intelligible fashion and written in standard English?

Reviewer #1: Yes

Reviewer #2: Yes

Reviewer #3: Yes

5. Review Comments to the Author

Reviewer #1: Dear study authors

Thank you for the opportunity to review this manuscript. You have undertaken a study to describe the prevalence and characteristics of self-reported baseline musculoskeletal in a cohort of European music students participating in the RISMUS study.

Thank you for your very interesting and thoughtful manuscript. It was well written and a pleasure to read. I have a few small queries, mainly for clarification. Thank you for the attention to detail in your work.

1. Introduction: It would be useful to include a sentence on your initial hypothesis.

2. Table 1: Could you please add what region each country belongs to. It would assist with interpretation of the results.

3. Page 4, line 112: you have a typo here for 997, it should read nine hundred and ninety seven, or just type the number.

4. I’m a little unclear about why you excluded people with a positive history of neurological, or psychological disorders given you are looking at injuries and psychological health (page 5). I assume you mean chronic or severe health conditions rather than acute or less severe symptoms? For example, you have mentioned tingling in your description of injuries, which I would interpret as a neurological condition. The same with psychological health, what was your cutoff and how did you classify it. Could you please include a few examples here to clarify your exclusions.

5. Methods: (page 6), Self-report health item: could you please include the anchor points here (e.g. 5 points, poor to excellent)

6. Statistical significance: I am a little concerned about the “reaching significance” of <0.1 described. You may wish to remove this, I do not think it adds anything to your message, especially given the size of your cohort.

7. Page 16, line 415, You appear to have missed the Barret reference no. here.

8. Page 17, line 450. Seems to need a slight re-wording. Perhaps …and thus assessing the probability…

9. Figure 1: you are missing a ) on the n=8 (not meeting inclusion criteria)

Reviewer #2: This is a valuable study and I was delighted to see this come in for review. This type of information is very much needed. The conclusions are clear and well justified and the information presented in a way that is easily understood by the reader. I have much praise.

Just a couple of tiny things to consider:

There are a few typos in the manuscript (eg p3, line 72 is it "has" instead of "had"?; p3, line 79 is it "Therefore, there is a need. . "; p3 line 94 is it "analysis" or "analyses" at end of line (I'm assuming it's plural?)

I personally would like to have seen a table that summarizes the numbers of participants according to the six year group levels, and the six types of instruments. This would allow me to see that there were sufficient respondents in each cell.

Also, p8, lines 217 - this is really methodology not results, so this section would fit better, I think, within one of the previous sections where the methodology and procedure is detailed.

This is a great study though and one that I look forward to reading when it comes out in print.

Reviewer #3: The authors report a cross-sectional study on music students' musculoskeletal problems across European countries including a sample of 850 participants. They ask about the prevalence and associated factors contributing to health conditions and observe that a majority of respondents (560) reported problems. Some contributing factors are identified and discussed. They conclude that this study is new.

My overall impression is an intersting piece of work. However, I can see (lots of) scope for refinement and improvement before publication could be recommended. The conclusions are off-topic, which is I rated them as partially convincing. However, this could be fixed, but all sections need to be addressed.

Abstract

I am not sure whether musculoskeletal disorders need to be a priori and categorically be playing-related. Students could fall off a bicycle or injure themselves during sports to report problems. I can understand that those are less interesting, but they still happen and perhaps put into relation to playing-related problems. The second large group of conditions which affects musicians are mental health issues (e.g. performance anxiety). Physical and psychological health problems seem to interact. https://pubmed.ncbi.nlm.nih.gov/29600306/ The results suggest that perceived exertion could play a role. Therefore, I wonder whether psychological factors should be also addressed as key dependent measures, if such variables were recorded.

Conclusions: The authors should conclude about their findings rather than just saying they did a heroic job. And it is not true that this is the first study of this kind:

Kreutz, G., Ginsborg, J., & Williamon, A. (2008). Music students' health problems and health-promoting behaviours. Medical Problems of Performing Artists, 23(1), 3-11.

See also

https://pubmed.ncbi.nlm.nih.gov/30204822/

https://pubmed.ncbi.nlm.nih.gov/24925174/

https://pubmed.ncbi.nlm.nih.gov/30061850/

https://pubmed.ncbi.nlm.nih.gov/20795333/

https://pubmed.ncbi.nlm.nih.gov/31130887/

https://pubmed.ncbi.nlm.nih.gov/29066983/

https://pubmed.ncbi.nlm.nih.gov/24647455/

https://pubmed.ncbi.nlm.nih.gov/24925172/

https://pubmed.ncbi.nlm.nih.gov/27942697/

Recently, for example, unbalanced posture patterns have been identified as a potential cause of problems.

https://pubmed.ncbi.nlm.nih.gov/32655447/

I believe that reflecting this broader literature on music students and health (focusing on conservatoire students) could tremendously enrich the current paper and expand its base in the introduction as well as enrich the conclusions that are drawn from the data. For example, how do the prevalence findings by Kreutz and colleagues (2008) relate to the current data, which did not include students from the UK, by the way? I do not wish to impose new research questions to these authors, but instead I believe that answering those questions should entail a discussion of these related findings.

I find a review of the literature that seemed to have escaped the authors' attention, rather important as background to the present study. I am surprised that the study protocol seems to have been approved without the demand to review the literature more thoroughly, what could have surfaced in the present manuscript

In brief, the authors should write a conclusion in the abstract that reflects the implications of their findings. Having conducted a study, whether or not it is genuinely a new study, is not an implication at all.

Introduction

I invite the authors to explore and review the literature referenced above. I believe that many, if not most study will be relevant in the present context to expand the introduction.

The authors should critically reflect on Zaza's categorizition instead of simply accepting it without hesitation. Specifically, what does "playing-related" entail, and what does it not entail? This is not just a matter of definition, but, as we are social scientists, a matter of developing an appropriate model that could predict the prevalence of musculoskeletal problems. Counting those and putting them in a basket is not what the authors intend to do, I believe. And given that the authors succeed in identifying associated factors, this information could be used to develop a (simple) model as a starting point. I am not convinced that setting up a project can act as a replacement for such a model. Instead of the last paragraph of the Introduction, a section header "Aims, Research Questions, and Hypotheses" is needed to guide the reader through the research interests of this group. Setting up a project is just a means, but does not give a clue what the authors want to learn.

Materials and Methods

- please refrain from overstatements such as "for the first time". The authors have not thoroughly reviewed the literature and are in no position for such sweeping claims. Moreover, of what value are such assertions in this section?

Table 1: It would be more informative to learn the percentage of participants relative to the total student population at each conservatiore. It may or may not turn out that the smaller conservatoires contribute a relatively higher percentage of participant. This could be a source of bias as students at larger (perhaps) more prestiguous institutions show less interest.

What are "assessment measures"? (p6, top). Instead the authors should present dependent and independent measures of their study. Those are important. It appears inappropriate to put in a reference rather than stating (in brief) what the measures are.

The measurement instruments could be reported in an Appendix. They should also present more specifics about which variables were used as demographic and musical background, playing-related problems, or variables that were then identified as associated factors. In other words, please categorize your sets of variables such that reader can gain a better overview. A Table would be very helpful.

Statistical Analysis

The first para is difficult to understand. The first two lines relate to categorical, the third sentence to continuous variables? Please be more clear. Did you consider confidence intervals to represent continuous measures?

"In addition, a multivariable analysis was conducted with an explorative aim in order to assess, at a multivariate level ..." this is a tautology.

Results

Again, you could well report how many students relative to the approximate total student population participated.

There seems to be a fair amout of significant psychological factors in the regression models. I think that this should be better reflected in the Abstract.

Discussion

The first para reports results. But the Discussion should address implications of those results, using numbers only to a minimum. I think that it is important to formulate the aims, research questions and hypotheses more clearly at the end of the intro as an independent section just because those can be addressed in the Discussion.

Interpreting regression models at a formal level distinguishes between predictor and criterion variables. It seems correct to me to consider those relationships between variables as associations. Nevertheless, I would find it appropriate on the basis of those models to talk about the extent to which some of the independent variables predict health problems. The implication is, that if it is playing-related, playing musical instruments must be assumed as a cause. But the direction of causality is not so clear. In that sense, psychological factors such as fatigue or perfectionism might predict playing-related health issues. On the other hand, musculoskeletal problems may cause greater fatigue etc. Readers might benefit from a differentiated discussion. If the authors proposed a model to base their assumptions upon, it would be easier to discuss findings in relation to that model.

I think that one point for Discussion might also be what student behaviours might contribute or not to better health by referencint studies such as Kreutz, G., Ginsborg, J., & Williamon, A. (2009). Health-promoting behaviours in conservatoire students. Psychology of Music, 37(1), 47-60.

The Discussion should at least briefly address potential mechanisms which drive the observed associations.

Limitations

"Despite the novelty and original approach ... " Avoid such phrasing as it has no meaning to the content of this study.

Limitations could also address the need for more elaborate models that entail mechanisms and moderators in the identified associations. How about students engaging in health-prevention programs? Is there reason to believe that they could benefit from reduced health problems? How about aerobic fitness, mental health programs etc.? In other words, one concrete limitation is that information of individual and/or institution level health-prevention needs to be more fully addressed to better understand the current findings.

Currently, due to the corona pandemic, the quality of education appears to deteriorate. Will this bring larger health problems in the future? Personally, I believe that music students may be exposed to greater health risks through distance teaching. On the other hand, a decline could also be expected as practice intensity could be limited by lack of availability of practice rooms for some instrumentalists.

Conclusions

"The present study reports a substantial part of the findings from the baseline examination of the longitudinal research project

491 RISMUS..." - This is not a conclusion.

"offers valuable insights" avoid such contentless phrasing. Readers may judge themselves what is of value to them. This is not your job.

Why does the conclusion repeat the first para of the Discussion. Neither here nor there is the content appropriate.

"statistical approaches has not been conducted before among European music students at different stages of their education." - That is not true and authors are advised to refrain from such phrasing.

"...primary prevention, including raising awareness within the musical and scientific

502 community, is important for the development of successful interventions and programs..." - The authors should appreciate the efforts documented in an extensive research literature first before proposing such sweeping demands.

6. PLOS authors have the option to publish the peer review history of their article (what does this mean?). If published, this will include your full peer review and any attached files.

Reviewer #1: No

Reviewer #2: **Yes: **Gary E McPherson

Reviewer #3: No

---

## [Author Response · Author response to Decision Letter 0]

26 Oct 2020

We would like to sincerely thank the reviewers for the very careful review of our paper and for the valuable comments, corrections and constructive criticism, which were of great help in revising the manuscript and helped to significantly improve the quality of the manuscript. 

We believe that the revised manuscript has been systematically improved accordingly, with new information and additional interpretations. 

Below, you will find a point-by-point description of how each point has been addressed in the manuscript.

[REVIEWER #1]

Dear study authors

Thank you for the opportunity to review this manuscript. You have undertaken a study to describe the prevalence and characteristics of self-reported baseline musculoskeletal in a cohort of European music students participating in the RISMUS study.

Thank you for your very interesting and thoughtful manuscript. It was well written and a pleasure to read. I have a few small queries, mainly for clarification. Thank you for the attention to detail in your work.

We appreciate the positive feedback, as well as your accurate and careful reading.

1. Introduction: It would be useful to include a sentence on your initial hypothesis.

We fully agree with this comment and following your recommendations, we have now added a sentence containing our initial hypothesis in the introduction. 

[Page 4, line 101-104]: “Our hypothesis was that there is a higher prevalence of PRMDs among university-level students in comparison with pre-college students (i.e. transition between pre-college and university-level) possibly due to the assumption that the exposure of playing-related activities is progressively demanding throughout their training.”

2. Table 1: Could you please add what region each country belongs to. It would assist with interpretation of the results.

Thank you for your important comment. We apologise that the information was not clear enough. The information contained within Table 1 is related only to the locations of the involved conservatories with the number of students who completed the web-based questionnaire. The nationality, classified for convenience in regions in Table 3 (Descriptive statistics of demographic variables) was an important variable (i.e. associated factor) but did not necessarily correspond to the locations of universities. Unfortunately, we were not allowed to match nationality and university’s membership for ethical reasons and thus we are not able to add the information you have kindly suggested to add. 

3. Page 4, line 112: you have a typo here for 997, it should read nine hundred and ninety seven, or just type the number.

Thank you for bringing this typo to our attention. The word “ninety” has been replaced by “nine [Page 4, line 122].

4. I’m a little unclear about why you excluded people with a positive history of neurological, or psychological disorders given you are looking at injuries and psychological health (page 5). I assume you mean chronic or severe health conditions rather than acute or less severe symptoms? For example, you have mentioned tingling in your description of injuries, which I would interpret as a neurological condition. The same with psychological health, what was your cutoff and how did you classify it. Could you please include a few examples here to clarify your exclusions.

Thank you for your important comment and please accept our apologies for not being clear enough in our exclusion criteria. 

We fully agree with your comment and following your recommendations, we have now revised the exclusion criteria.

[Page 6, line 130-131]: “…positive history of chronic and highly disabling neurological and/or rheumatic and/or psychological conditions in the last 12 months”

We excluded chronic and highly disabling mental or physical conditions (i.e. focal dystonia, fibromyalgia syndrome, rheumatoid arthritis or borderline personality disorder) which might prevent students from playing and practising. Nonetheless, we’d like to reiterate that this study comprises part of a larger longitudinal study that involves cohorts of music students who, according to their health condition, were able to play and practice. 

5. Methods: (page 6), Self-report health item: could you please include the anchor points here (e.g. 5 points, poor to excellent)

Thank you for your valuable comment. Following your suggestion, we inserted the anchor points for self-rated health item.

[Page 6, line 151]: “…, answered on a five-point scale from excellent to poor…”

6. Statistical significance: I am a little concerned about the “reaching significance” of <0.1 described. You may wish to remove this, I do not think it adds anything to your message, especially given the size of your cohort.

Thank you for bringing this to our attention. Thanks to your comment, we have realised that p-values were wrongly reported and have now been corrected.

[Page 13, line 300; Page 14, line 329]: “*** p<0.001, ** p<0.01, * p<0.05”

[Page 13, line 308-310]: “Nationality, academic level, perfectionism, fatigue, years of practice and perceived exertion after 45 minutes of practice without breaks (χ2 (df, 2 to 10) = 10.4 to 49.5; p<0.001), as well as psychological distress (χ 2 (df, 2) = 8.4; p<0.01) were related significantly with MSK status (NoMSK, PRMD and MSK).”

7. Page 16, line 415, You appear to have missed the Barret reference no. here.

Thank you for highlighting this. The reference number has now been included, as kindly pointed out [Page 17, line 428].

8. Page 17, line 450. Seems to need a slight re-wording. Perhaps …and thus assessing the probability…

Thank you for the suggestion. The sentence has been revised, as suggested [Page 18, line 483]. 

9. Figure 1: you are missing a ) on the n=8 (not meeting inclusion criteria)

Thank you for your careful reading and for highlighting this. We have now added the parenthesis in Figure 1.

REVIEWER #2 (GARY MC PHERSON)

This is a valuable study and I was delighted to see this come in for review. This type of information is very much needed. The conclusions are clear and well justified and the information presented in a way that is easily understood by the reader. I have much praise.

We would like to frankly thank Prof Mc Pherson for the positive feedback and valuable suggestions. 

Just a couple of tiny things to consider:

There are a few typos in the manuscript (eg p3, line 72 is it "has" instead of "had"?; p3, line 79 is it "Therefore, there is a need. . "; p3 line 94 is it "analysis" or "analyses" at end of line (I'm assuming it's plural?)

Thank you for highlighting this. We found your comments extremely helpful and have revised accordingly. 

[Page 3, line 75]:” …has made comparison…” 

[Page 3, line 84]:”Therefore, there is a need to…”

In addition, the word “analysis” has been replaced by “analyses” [Page 4, line 107].

I personally would like to have seen a table that summarizes the numbers of participants according to the six year group levels, and the six types of instruments. This would allow me to see that there were sufficient respondents in each cell.

Thank you for your important comment. Following your suggestion, we inserted a table containing the number of music students according to the six-year group levels and the six types of instrument to assist the reader with interpretation of the results.

[Page 11, line 258-259]: “Table 6 includes the distribution of participants, according to the six-year group levels and the six types of instruments’ classification.”

Also, p8, lines 217 - this is really methodology not results, so this section would fit better, I think, within one of the previous sections where the methodology and procedure is detailed.

Thank you for your valuable comment, which made us reflect on this important aspect. We have checked the STROBE checklist for cross-sectional studies https://www.strobe-statement.org/index.php?id=available-checklists, where it is recommended (but not stipulated) to report this information (i.e. Examination of for eligibility, inclusion in the study, use of a flow diagram) in the results’ section (section of the STROBE cross-sectional studies: 13c). 

Nonetheless, if you would prefer in this manuscript, for us to move this paragraph [Page 9, line 227-230] to the methods’ section, we will be delighted to do so.

This is a great study though and one that I look forward to reading when it comes out in print.

Thank you for your appreciation and encouragement! They are very much appreciated.

REVIEWER #3

The authors report a cross-sectional study on music students' musculoskeletal problems across European countries including a sample of 850 participants. They ask about the prevalence and associated factors contributing to health conditions and observe that a majority of respondents (560) reported problems. Some contributing factors are identified and discussed. They conclude that this study is new.

My overall impression is an intersting piece of work. However, I can see (lots of) scope for refinement and improvement before publication could be recommended. The conclusions are off-topic, which is I rated them as partially convincing. However, this could be fixed, but all sections need to be addressed.

We would like to thank you for taking time and effort necessary to review our manuscript. We sincerely appreciate your valuable comments and we are grateful for your suggestions and thoughts.

Abstract

I am not sure whether musculoskeletal disorders need to be a priori and categorically be playing-related. Students could fall off a bicycle or injure themselves during sports to report problems. I can understand that those are less interesting, but they still happen and perhaps put into relation to playing-related problems. 

The second large group of conditions which affects musicians are mental health issues (e.g. performance anxiety). Physical and psychological health problems seem to interact. https://pubmed.ncbi.nlm.nih.gov/29600306/

The results suggest that perceived exertion could play a role. Therefore, I wonder whether psychological factors should be also addressed as key dependent measures, if such variables were recorded.

Thank you for your important comment. 

It is entirely plausible that amongst the challenges of writing towards a word-limit, we would not been entirely successful in describing our intentions, and for which we apologise for causing any confusion. If we may clarify, we are not interested solely in investigating music students’ musculoskeletal (MSK) disorders, but instead, we are also focusing on the impact of how existing musculoskeletal conditions and symptoms might influence the way in which music can be played, practised and performed. Although the definition of PRMDs does not provide a causality of the disorder (i.e. the disorder is the result of playing the instrument), and we would readily concede that there are inevitable difficulties in selecting the scope of this field of study (to which you allude), we would suggest politely that it could be seen as being the best definition currently available and it has been deployed often in the contemporary literature (Ackermann and Driscoll, 2010 21120266; Ackermann et al., 2011 22045531; Ackermann et al., 2012 23247873; Ajidahun and Philips, 2013 23752284; Arnason et al., 2014 24925174; Baadjou et al., 2016 27138935; Baadjou et al., 2018 30085148; Berque et al., 2016 27281378; Chan et al., 2014a 24213243; Chan et al., 2014b 25433253; Kaufman- Cohen and Ratzon, 2011 21273187; Kenny et al., 2016 26966957; Kochem and Silva, 2018 30077421; Möller et al., 2018 30204820; Sousa et al., 2015 26343102; Sousa et al., 2017 28555556; Steinmetz et al., 2012 23138678; Steinmetz et al., 2015 24389813). In regard to the latter point, we would therefore prefer to use Zaza et al.’s definition of PRMD in order to facilitate the interrogation of the literature and to offer comparability with data from studies already published in this area. The authors had felt that the last point was particularly important because, according to recent systematic reviews, the extensive heterogeneity of types of methods and definitions already evident amongst the literature has made synthesis of the evidence limited if not impossible (Kok et al., 2016 26563718; Rotter et al., 2019 31482285). 

In addition, we sought to include other MSK disorders in the analysis and to distinguish them from PRMDs in accordance with recommendations from the performing arts medicine field (Baadjou et al., 2016 27138935; Kok et al., 2016 26563718). We think that the distinction between the two categories was very important because it allowed descriptive contrast amongst factors associated with the general presence of MSK conditions and factors specifically related to PRMDs. 

Nonetheless, we very much appreciate your helpful suggestion of a wider remit for factors affecting musculoskeletal disorders, as it’s something with which the authors have already wrestled for an ideal solution. In the future, we would gladly wish to consider the much greater scope associated with musculoskeletal disorders in general. However, in this instance and for the reasons that we have offered, we have been persuaded that on balance, it would be reasonable to offer a more focal consideration on musculoskeletal conditions and symptoms that affect the playing of music.

In relation to mental health, we agree wholeheartedly with your insightful comments that it could be also a problem concerning musicians. With this possibility in mind, we did include a global measure of psychological distress based on questions about anxiety and depressive symptoms (i.e. Kessler Psychological Distress Scale K10 developed within the Harvard Medical School, Boston, USA) to verify the interaction between psychological distress and MSK status [Page 2, line 36].

Nonetheless, in this first instance of exploration, our study was designed to focus specifically on musicians MSK disorders and PRMDs. If at all possible, we are hoping that it would be seen as being reasonable to keep the focus and coherence intact. In future studies however, it may be entirely relevant and desirable to evaluate the association between performance anxiety (as a dependent variable) and physical health factors, in order to gain greater insight into the mechanisms and personal significance of MSK disorders among musicians in relation to the performance. 

Conclusions: The authors should conclude about their findings rather than just saying they did a heroic job. And it is not true that this is the first study of this kind:

Kreutz, G., Ginsborg, J., & Williamon, A. (2008). Music students' health problems and health-promoting behaviours. Medical Problems of Performing Artists, 23(1), 3-11.

See also

https://pubmed.ncbi.nlm.nih.gov/30204822/

https://pubmed.ncbi.nlm.nih.gov/24925174/

https://pubmed.ncbi.nlm.nih.gov/30061850/

https://pubmed.ncbi.nlm.nih.gov/20795333/

https://pubmed.ncbi.nlm.nih.gov/31130887/

https://pubmed.ncbi.nlm.nih.gov/29066983/

https://pubmed.ncbi.nlm.nih.gov/24647455/

https://pubmed.ncbi.nlm.nih.gov/24925172/

https://pubmed.ncbi.nlm.nih.gov/27942697/

Recently, for example, unbalanced posture patterns have been identified as a potential cause of problems.

https://pubmed.ncbi.nlm.nih.gov/32655447/

I believe that reflecting this broader literature on music students and health (focusing on conservatoire students) could tremendously enrich the current paper and expand its base in the introduction as well as enrich the conclusions that are drawn from the data. For example, how do the prevalence findings by Kreutz and colleagues (2008) relate to the current data, which did not include students from the UK, by the way? I do not wish to impose new research questions to these authors, but instead I believe that answering those questions should entail a discussion of these related findings.

I find a review of the literature that seemed to have escaped the authors' attention, rather important as background to the present study. I am surprised that the study protocol seems to have been approved without the demand to review the literature more thoroughly, what could have surfaced in the present manuscript

In brief, the authors should write a conclusion in the abstract that reflects the implications of their findings. Having conducted a study, whether or not it is genuinely a new study, is not an implication at all.

The authors are very grateful for this insightful selection of comments.

In regard to the abstract’s conclusions, we apologise if there was a sense inadvertently of self-congratulation: That wasn’t our intention in any way. The authors are very grateful to the reviewer for highlighting a selection of additional antecedent studies contributing to this field of study.

In that context, and as we’ve alluded to previously, our reason for maintaining that our study is new and original may not have been explained with sufficient clarity. Nevertheless, according to our knowledge, we think that it’s reasonable to conclude specifically that there are currently no large-scale studies on music students who have included several university centres/conservatoires in Europe and at the same time investigating prevalence and associated factors for both PRMDs and musculoskeletal disorders at different levels of study (from pre-college to Masters levels). We were routinely aware that there are several studies in the literature (including a selection of those that you’d kindly listed) that had considered the prevalence of musculoskeletal disorders of music students but, according to our knowledge, they are not multicentre studies taking into consideration the important distinction between PRMDs and generic MSK disorders. 

We have been pleased nevertheless to incorporate your advice about the manuscript in this context and have revised the conclusions accordingly. We have now hopefully stated the novelty of our study in a more precise way and consistently with the aims.

[Page 2, line 41-45]: “According to the authors’ knowledge, a large-scale multicentre study investigating prevalence and associated factors for PRMDs among music students at different stages of their education (from Pre-college to Masters levels) has not been conducted before. The high prevalence of PRMDs among music students, especially those studying at university-level, has been confirmed in this study and associated factors have been identified, highlighting the need for relevant targeted interventions as well as effective prevention and treatment strategies.”

In order to be consistent, the title has also been changed. In addition, the name of the project has been added as well. 

[Page 1, line 1-3]: “Prevalence and associated factors of playing-related musculoskeletal disorders among music students in Europe. Baseline findings from the Risk of Music Students (RISMUS) longitudinal multicentre study”

The authors have taken the liberty of discussing the contribution of Kreutz et al. briefly within the next section of our responses to your comments.

The work of Kreutz et al. is very interesting as it focuses on the prevalence of both musculoskeletal and non-musculoskeletal health problems in music students and their relationship to perceived practice and performance quality as well as to students’ self-reported health-promoting behaviours. However, we would respectfully suggest that our study is distinct from Kreutz et al.’s in several aspects, as detailed below:

1. Although Kreutz and colleagues focused on music students, neither the age range of participants (in fact, this study has been excluded by two recent systematic reviews – Kok et al., 2016 26563718; Rotter et al., 2019 31482285), nor the academic level are stated in the method section. Instead, we have focused our study on the prevalence’s difference between different academic levels (i.e. Pre-college and university-level).

2. In the study of Kreutz and colleagues, musculoskeletal and non-musculoskeletal problems were assessed in relation to different groups of instruments. However, the classification of instruments used within the study was determined according to the “instrument families” (i.e. keyboard instruments; strings; woodwinds; brass; plucked instruments; percussion; voice; composition). Since our study is mainly focused on MSK status, we think that participants should be grouped according to playing posture and the elevation of the arm as a recognised risk factor for the development of MSK disorders (Nyman et al., 2007 17427201; Kok et al., 2017 28282473). 

3. In the study of Kreutz et al., the second and third research questions/aims were related to the association between musculoskeletal and non-musculoskeletal problems and perceived practice and performance quality, as well as health-promoting behaviours. Although these aspects are interesting and inspiring, in this instance and for the reasons that we have offered previously in our responses to your comments, we have used the definition of Zaza to investigate the interference of musculoskeletal disorders on music students’ playing ability. In addition, we are grateful to you for your comments in this context and concur fully with your arguments that health-promoting behaviours are important. Having been very aware already of the critical role of behaviours, we would wish to consider this important aspect in future studies to gain greater insight into strategies toward health promotion that music students might adopt to protect them from the effects of the development of MSK disorders and PRMDs. 

However, while the authors would always be open to considering expert advice on these matters, we feel somewhat constrained currently as we have already received a favourable ethical approval for the project as a whole using the current definition of Zaza et al. and the study’s protocol with this definition has been already published. Although we understand the concerns and issues with this definition, as you might imagine, we are a bit reluctant to change the terms and criteria used for recruitment that have been approved and are available in the public domain. 

In relation to the studies you have kindly suggested, we have been delighted to now include some of them that enhance our manuscript, and for which we thank you. 

[Page 3, line 78-80]: “Similarly, although there is a growing literature regarding MSK among music students (10, 11, 25, 28, 32-39) and preventive courses as well as short-term health education programs have proliferated during the last twenty years (40-45) …”

Having undertaken a careful review of the antecedent and contemporary literature, we’re still of the opinion that the contemporary literature in particular offers a large heterogeneity of methods amongst small samples that limit generalisations and meta-analytical synthesis of the evidence. We were reassured because these sentiments have also been reinforced recently by Rotter and colleagues (2019) within their systematic review. In light of the latter, it would seem that studies with larger sample-sizes, involving different institutions and countries, as well as students at different levels of training are needed. In this regard, we were hopeful that our study involving a multicentre research, a relatively large sample and with different levels of training sought correctly in some small way to build on the accomplishments of those already contributing to knowledge in the topic area.

Unfortunately, some of the studies you have kindly mentioned have not been inserted because we felt that they were somewhat tangential to our study and instead they focused mainly on wellbeing (Antonini et al., 2019) or had not focused on musculoskeletal conditions (Araujo et al., 2017; Spahn et al., 2014) or included music students under 16 years of age (Romero et al., 2016).

Similarly, in relation to unbalanced posture patterns, we would like to thank you very much for the suggestion of this reference, about which we did not know because was published after our submission. In this instance, Nusseck and Spahn have investigated postural stability and balance in musicians compared with a control group. Although this approach is very interesting and inspiring to us, unfortunately, in their results, no significant differences were found amongst groups for problems of the musculoskeletal system. As such, unbalanced posture patterns had not been identified as a potential cause of musculoskeletal problems. Furthermore, it was not clear how patterns of unbalanced posture had been assessed, as the description of its outcome measure was missing within the methods’ section.

Introduction

I invite the authors to explore and review the literature referenced above. I believe that many, if not most study will be relevant in the present context to expand the introduction.

The authors should critically reflect on Zaza's categorizition instead of simply accepting it without hesitation. Specifically, what does "playing-related" entail, and what does it not entail? This is not just a matter of definition, but, as we are social scientists, a matter of developing an appropriate model that could predict the prevalence of musculoskeletal problems. Counting those and putting them in a basket is not what the authors intend to do, I believe. And given that the authors succeed in identifying associated factors, this information could be used to develop a (simple) model as a starting point. I am not convinced that setting up a project can act as a replacement for such a model. Instead of the last paragraph of the Introduction, a section header "Aims, Research Questions, and Hypotheses" is needed to guide the reader through the research interests of this group. Setting up a project is just a means, but does not give a clue what the authors want to learn.

Once again, the authors are very grateful to you for your comments above specifically focusing on the manuscript’s introduction and to which we’ve paid particular attention. We should be grateful if you’d consider the following responses alongside those offered earlier which relate to issues of definition and a conceptual framework for the research. 

We’d like to reiterate that this study comprises part of a larger longitudinal study, which aims to identify factors associated with increased risk of PRMDs in music students. We’d considered carefully that a robust way in which to search for risk factors was to conduct a prospective, longitudinal study, with a large sample size and an a priori-defined model based on current literature (Hayden et al., 2013 23420236). This model should adequately represent and encompass the population (e.g. different academic levels and different instruments, as well as different countries). 

Among the relevant literature, which has been deeply explored on our part, the definition of exposure of most studies was often insufficient (Rotter et al., 2019 31482285). As authors, we are attempting to be as thorough and diligent as possible and we would wish to follow the recommendations of latest systematic reviews (Baadjou et al., 2016 27138935; Kok et al., 2016 26563718; Rotter et al., 2019 31482285), which we think have been conducted in a respectable and careful way. In addition, the authors deemed it important to provide clear definitions and valid outcome measurements of risk factors in order to follow current methodological requirements. In fact, according to a recent systematic review, the body of evidence concerning prevalence, risk factors and effectiveness of the prevention or treatment of MSK disorders among musicians is still missing, mainly due to methodological concerns (Rotter et al., 2019 31482285). 

PRMD is a collective term that was defined by Zaza et al. (1998 10075243) as “any pain, weakness, numbness, tingling or other symptoms that interfere with the ability to play your instrument at the level you are accustomed to”. This definition has been already used in several studies with music students (Baadjou et al., 2016 27138935; Baadjou et al., 2018 30085148; Arnason et al., 2014 24925174; Möller et al., 2018 30204820; Steinmetz et al., 2012 23138678; Zaza et al., 1998 10075243). One of those is indeed of Zaza et al., who collected data from 281 classically trained professional musicians and university music students. 

We understand and agree that this definition has many limitations and currently, a gold standard definition for musculoskeletal disorders related to musicians’ playing activity does not exist. However, in order to avoid continuing to nourish heterogeneity in this field of research, the authors felt that it would be both a useful and reasonable strategy to deploy the most recognised one. Therefore, we have attempted to use the best possible current definition, but at the same time differentiating the characteristics associated with these disorders from the more generic ones (MSK) in this explorative paper. It is not able to address all aspects, but the definition used offers the reasonable approach to a step-forward compared to what might have conducted before. 

Nonetheless, following your valuable suggestion a new paragraph containing the aims of the study has been inserted.

[Page 4, line 98-107]: “Aims. The purpose of the present study was to examine the prevalence of PRMDs in a large-scale study population of music students enrolled in different pan-European music institutions at baseline of the RISMUS project, in order to characterise the study population at different levels of training (i.e. university-level students and Pre-college students). Our hypothesis was that there is a higher prevalence of PRMDs among university-level students in comparison with pre-college students (i.e. transition between pre-college and university-level) possibly due to the assumption that the exposure of playing-related activities is progressively demanding throughout their training. A further goal was to begin to identify variables that might be associated with the self-reported presence of PRMDs among music students. Specifically, an approach involving multivariable modelling might offer preliminary explorative and novel insights of the baseline findings to be further verified within the longitudinal analyses.”

Materials and Methods

- please refrain from overstatements such as "for the first time". The authors have not thoroughly reviewed the literature and are in no position for such sweeping claims. Moreover, of what value are such assertions in this section?

Thank you very much for this comment.

As we’ve alluded to previously within our responses to you, we have now hopefully stated the novelty of our study in a more precise way and consistently with the aims ([Page 2, line 41-45]), and will refrain from reiterating this aspect unnecessarily, especially within a ‘materials and methods’ section, where of course, it’s entirely superfluous.

In addition, we have now moved the information of the longitudinal investigation “RISMUS” in the introduction, without repeating it in the methods’ section. 

[Page 3, line 89-92]: “The Risk of Music Students (RISMUS) research project was set up in 2018 to characterise clinical features of a large sample of students from pan-European music institutions and to longitudinally identify factors associated with increased and evolving risk of playing-related musculoskeletal disorders during their professional training (32). This 12-months longitudinal multicentre investigation has evolved to incorporate recommendations within the current literature…”

Table 1: It would be more informative to learn the percentage of participants relative to the total student population at each conservatiore. It may or may not turn out that the smaller conservatoires contribute a relatively higher percentage of participant. This could be a source of bias as students at larger (perhaps) more prestiguous institutions show less interest.

Thank you for your suggestion, with which we fully agree. Unfortunately, responses from all the institutions have not yet been forthcoming and without the accurate data about student numbers and permission to publish, the authors are constrained in this respect for this baseline manuscript. Should the data become available prior to publication, then we’d be delighted to include the relevant figures at the earliest possible opportunity.

Nonetheless, we have mentioned this aspect as a limitation. 

[Page 19, line 520-522]: “Furthermore, the authors cannot exclude a potential sampling bias as the information concerning the number of students enrolled in each school participating in the study is not available because it consists of confidential data, without a formal permission to publish. “

What are "assessment measures"? (p6, top). Instead the authors should present dependent and independent measures of their study. Those are important. It appears inappropriate to put in a reference rather than stating (in brief) what the measures are.

The measurement instruments could be reported in an Appendix. They should also present more specifics about which variables were used as demographic and musical background, playing-related problems, or variables that were then identified as associated factors. In other words, please categorize your sets of variables such that reader can gain a better overview. A Table would be very helpful.

Thank you for your comment. “Assessment” has been replaced by “Outcome” [Page 6, line 137] 

We have actually reported in brief what the measures concern together with the psychometric properties [Page 6-7, line 150-167] to offer the reader a better understanding of the outcome measures used. 

In relation to the measurement instruments, it is already available as an appendix of the published protocol [Page 6, line 138-139]: “…which are available in the published protocol”

Following your kind suggestion, the clarification of dependent variable (MSK status) has been added and a table that hopefully describes the variables in a more clear and specific way has been inserted to assist the reader with interpretation of the results. 

[Page 7, line 186-188]: “Bivariate analysis was used to identify associations between the dependent variable MSK status and the covariates (i.e. demographic variables, as well as variables associated with health-related status and those associated with the playing of musical instruments) (see Table 2)”.

Statistical Analysis

The first para is difficult to understand. The first two lines relate to categorical, the third sentence to continuous variables? Please be more clear. Did you consider confidence intervals to represent continuous measures?

Thank you for highlighting this please accept our apologies for not being clear enough. The paragraph has now been revised accordingly.

[Page 7, line 182-184]: “For categorical variables, absolute and relative frequency distributions were presented. For continuous variables, since the normality test showed that all the variables considered were non-Gaussian, the median value and the range were used to summarise the variables.”

"In addition, a multivariable analysis was conducted with an explorative aim in order to assess, at a multivariate level ..." this is a tautology.

Thank you for pointing out this. “At multivariate level” has been deleted [Page 8, line 202].

“…a multivariable analysis was conducted with an explorative aim in order to assess...”

Results

Again, you could well report how many students relative to the approximate total student population participated.

There seems to be a fair amout of significant psychological factors in the regression models. I think that this should be better reflected in the Abstract.

Once again, the authors are very grateful to you for your comments above specifically focusing on the total number of students of each school that participated in the study. As alluded to previously in our responses to your query about Table 1, details of student numbers from all the institutions, although requested by the authors, have not yet been forthcoming. We reiterate here that without the accurate data about student numbers and permission to publish, the authors are constrained in this respect for this baseline manuscript. Should the data become available prior to publication, then we’d be delighted to include the relevant figures at the earliest possible opportunity.

Nonetheless, we have mentioned this aspect as a limitation. 

[Page 19, line 520-522]: “Furthermore, the authors cannot exclude a potential sampling bias as the information concerning the number of students enrolled in each school participating in the study is not available because it consists of confidential data, without a formal permission to publish.”

According to the bivariate analysis, psychological distress was related with MSK in general (PRMDs and non-PRMDs, with a higher median for MSK than PRMD or noMSK). We fully appreciate the need to reflect the scope of findings, especially in relation to psychological factors, which we agree are potentially very important. However, on this occasion, we’ve thought that it would be reasonable to report only factors in relation to PRMDs, in order to be consistent with the stated aims of the study.

Discussion

The first para reports results. But the Discussion should address implications of those results, using numbers only to a minimum. I think that it is important to formulate the aims, research questions and hypotheses more clearly at the end of the intro as an independent section just because those can be addressed in the Discussion.

We fully agree with this comment and following your recommendations, we have now deleted results from the first paragraph of the discussion section. 

[Page 15, line 359-368]: “This study focused on the prevalence of PRMDs in a large-scale study population of music students at different educational stages (i.e. university-level students and Pre-college students) and enrolled in different pan-European music institutions at baseline of the RISMUS project. Music students participating in this novel large-scale study involving 20 European countries self-reported a high prevalence of painful MSK conditions (65%), of those 48% self-reported PRMDs. 

A further goal was to begin to identify variables that might be associated with the self-reported presence of PRMDs among this population that ultimately would facilitate future longitudinal analyses. Results highlighted that coming from West Europe, being a first- or a second-year Masters student, having more years of experience and higher rates of perceived exertion after 45 minutes of practice without breaks were factors significantly associated with self-reported presence of PRMD. In this regard, the current study integrates novel and robust descriptive data with explorative and speculative analyses via relatively sophisticated statistical modelling for factors that may be associated with PRMDs (i.e. multinomial regression model).”

Interpreting regression models at a formal level distinguishes between predictor and criterion variables. It seems correct to me to consider those relationships between variables as associations. Nevertheless, I would find it appropriate on the basis of those models to talk about the extent to which some of the independent variables predict health problems. The implication is, that if it is playing-related, playing musical instruments must be assumed as a cause. But the direction of causality is not so clear. In that sense, psychological factors such as fatigue or perfectionism might predict playing-related health issues. On the other hand, musculoskeletal problems may cause greater fatigue etc. Readers might benefit from a differentiated discussion. If the authors proposed a model to base their assumptions upon, it would be easier to discuss findings in relation to that model.

Thank you very much for your important comment. 

As you might imagine, we have been very excited by the possibility through this research, as you’ve intimated, of being able eventually to identify factors that may be labelled as correlates, determinants, predictors, or indeed causal for provoking PRMDs. 

Nevertheless, for many of the reasons that we’ve already offered in these responses to your commentary, and especially in the context of this study being exploratory, we’ve opted to include terms such as “associated factors” and “associations” because on balance. We’d argue respectfully that cross-sectional studies cannot ultimately be considered appropriate to investigate the mechanisms of risk factors (Baadjou et al., 2016 27138935) or predictors. 

Even if it could be based on carefully constructed conceptual models and mediation-type analyses involving patterning of statistical correlation, the interpretation of associations from cross-sectional studies for direction of cause and effect it is extremely difficult to establish (Croft et al., 2001 11514090). As you’d be aware, this is because the presence of risk factors and the occurrence of an outcome (i.e. PRMDs) are being assessed simultaneously. 

For example, as you correctly suggested, fatigue may well be an important risk factor for the development of PRMDs, but it is equally possible that PRMDs might contribute to increased fatigue. We’d suggest that although it is plausible for there to be a robust conceptual and physiological framework to support the expectation of a linkage between the physical performance capability status characteristics of musicians and risks of developing PRMDs, there is currently neither sufficient scientific evidence for this linkage nor the basis on which clinical prevention of PRMD can be developed (Baadjou et al., 2016 27138935; Berque et al., 2016 27281378). 

Therefore, we might reasonable assume that these variables are related. Nevertheless, until the relative importance of fatigue compared to other candidate factors can be established, we would suggest that its utility may be compromised and even offer spurious indications for the mechanisms underpinning PRMDs. The latter would be suspected of being beyond the evidence from within this exploratory study. We would be very hopeful that our subsequent longitudinal analyses, which are in preparation, will have the capability of informing our understanding of these issues.

I think that one point for Discussion might also be what student behaviours might contribute or not to better health by referencint studies such as Kreutz, G., Ginsborg, J., & Williamon, A. (2009). Health-promoting behaviours in conservatoire students. Psychology of Music, 37(1), 47-60.

The Discussion should at least briefly address potential mechanisms which drive the observed associations.

Thank you for suggesting this important aspect, which has now been included in the discussion. 

[Page 16, 402-407]: “In addition, another possible reason for the reduction of PRMDs’ prevalence among Masters students at later stages could be that, although the literature reports that musicians engage poorly in health promoting behaviours (43, 67-69), courses and short-term health education programs have been recently developed to integrate useful insight from health professionals as well as knowledge from relevant health education settings (40, 42, 45). Students at later stages could have had the possibility to engage in these useful programs and reduce or treat their painful condition.”

[Page 17-18, line 439-460]: “For instance, a similar picture is provided by other studies that have investigated health perception among music students, who rated their health worse compared to an age-and sex matched group of students who did not play music and reported worst behaviour records of health responsibility (37, 43, 67-69). Similarly, Rickert et al. (78) reported an insufficient health awareness of injury among students playing the cello and Kreutz et al. (69) showed poor stress management, inadequate nutrition and low levels of health responsibility among music students, suggesting a consistent need for continuing to develop strategies to enhance health support as an essential aspect of conservatoire and music university education by for instance integrating it into students’ curricula and learning programs (43, 78, 79). During their professional training, music students should learn how to cope with physical and psychological demands with the help of preventive measures. Body-oriented courses (i.e. posture, strength and conditioning exercises) and relaxation techniques, as well as psychological programs for stress and wellbeing have been shown to have a preventive effect (42, 44, 45, 80). This indicates that better results on MSK conditions among music students could be obtained by addressing health awareness and attitudes to injury at the university or even at the Pre-college. Indeed, music universities represent the primary channel for the improvement of health awareness and the implementation of injury prevention initiatives, being an important gateway to the professional world (81). Therefore, strengthening attitudes and behaviours toward health music making will create a step change in educational and employment contexts, shaping future practice and addressing injury prevention to possibly avoid or at least reduce incidences of PRMDs. According to Rickert et al. (78) and Spahn et al. (80), health behaviours toward prevention may be easier to be addressed in the younger generation of musicians who may not already have such established habits. Preventive courses and health promotion among musicians should start already at the beginning of their musical training, with the objective to protect music students from PRMDs during their studies and to prepare them for the future professional demands. For instance, music students without a disorder at the beginning of their professional education would benefit of an increasing sensitization in health promotion and injury prevention. On the other hand, students already suffering from health concerns need to be informed about potential strategies to reduce symptoms (80).”

Limitations

"Despite the novelty and original approach ... " Avoid such phrasing as it has no meaning to the content of this study.

Limitations could also address the need for more elaborate models that entail mechanisms and moderators in the identified associations. How about students engaging in health-prevention programs? Is there reason to believe that they could benefit from reduced health problems? How about aerobic fitness, mental health programs etc.?

 In other words, one concrete limitation is that information of individual and/or institution level health-prevention needs to be more fully addressed to better understand the current findings.

Currently, due to the corona pandemic, the quality of education appears to deteriorate. Will this bring larger health problems in the future? Personally, I believe that music students may be exposed to greater health risks through distance teaching. On the other hand, a decline could also be expected as practice intensity could be limited by lack of availability of practice rooms for some instrumentalists.

“Despite the novelty and original approach…” has been deleted from the paragraph [Page 19, line 488].

In addition, following your suggestion, we have inserted the limitation regarding the impossibility to control for confounding factors. 

[Page 19, line 508-518]: “Moreover, another limitation consists of the impossibility to control information on the individual and/or the institutional level of behaviours or attitudes toward prevention. For instance, engaging in health-prevention programs could represent a potential confounder that might have affected our results. However, the web-based questionnaire includes questions on strategies to reduce any MSK conditions they may have had in the past and thanks to the replies of the two follow-ups we will have more information and we will be able to record this important aspect. In addition, the participatory level of physical activity has been monitored with the International Physical Activity Questionnaire (IPAQ), which is a well-known measure to offer data on health–related physical activity. Nonetheless, whereas it is important to consider individual health-promoting behaviours (68, 69), Perkins and colleagues (43) suggested that there is still the need to continue evaluating health behaviours and awareness among students and teachers inside music institutions, as well as environmental factors that might be perceived hampering or facilitating health and prevention. It is plausible to think that the environmental factors might be to some extent changed to accommodate research findings regarding the prevention of MSK conditions.”

Conclusions

"The present study reports a substantial part of the findings from the baseline examination of the longitudinal 

research project RISMUS..." - This is not a conclusion.

"offers valuable insights" avoid such contentless phrasing. Readers may judge themselves what is of value to them. This is not your job.

Why does the conclusion repeat the first para of the Discussion. Neither here nor there is the content appropriate.

Thank you for your feedback. The conclusions’ section has now been revised according to your very helpful suggestions. However, it includes a summary of the findings to be consistent with the aims, and the implications in the last sentences of the last paragraph. 

[Page 20, line 538-546]: “The high prevalence of PRMDs among music students, especially those studying at university-level, has been confirmed in this study and associated factors have been identified, highlighting the need for relevant targeted interventions as well as effective prevention and treatment strategies.

Although the results of this study should be interpreted with caution due to the cross-sectional and self-reported nature of the data, they reflect the findings from a relatively large-scale investigation involving multiple centres across Europe and importantly, students at different stages of their education (from Pre-college to Masters levels). These findings may contribute important adjunct findings to those from the antecedent literature facilitating effective approaches towards primary prevention of PRMDs and their associated burden among music students and professionals. They may usefully raise awareness further within the musical and scientific communities.”

"statistical approaches has not been conducted before among European music students at different stages of their education." - That is not true and authors are advised to refrain from such phrasing.

Thank you for your advice here.

We’ve omitted mention of our perceptions of what might have been attempted previously from the manuscript and as we’ve intimated from earlier, mention of the novelty of our study has been confined and delivered in a more precise way and consistently with the aims [Page 2, line 41-45].

"...primary prevention, including raising awareness within the musical and scientific community, is important for the development of successful interventions and programs..." - The authors should appreciate the efforts documented in an extensive research literature first before proposing such sweeping demands.

Once again, thank you very much for your helpful comments. 

We do sincerely appreciate what has been done in the extensive literature so far. As alluded to previously in our responses, while we have some misgivings about the methodological heterogeneity amongst some studies, we nevertheless recognise the importance of offering balanced critical evaluation of our findings in the context of the contemporary and historical literature. 

We are immensely grateful to you for the time in reviewing our manuscript and for your valuable assistance in improving its content with exceedingly useful comments, whether they differed from our thoughts or not: Your comments have helped our manuscript to become clearer and more readable.

As such, we’ve also taken your advice and modified the final aspects of the manuscript’s conclusions accordingly. 

[Page 20, line 543-546]: “These findings may contribute important adjunct findings to those from the antecedent literature facilitating effective approaches towards primary prevention of PRMDs and their associated burden among music students and professionals. They may usefully raise awareness further within the musical and scientific communities.”

---

## [Decision Letter · Decision Letter 1]

9 Nov 2020

Prevalence and associated factors of playing-related musculoskeletal disorders among music students in Europe. Baseline findings from the Risk of Music Students (RISMUS) longitudinal multicentre study

PONE-D-20-17856R1

Dear Dr. Cruder,

We’re pleased to inform you that your manuscript has been judged scientifically suitable for publication and will be formally accepted for publication once it meets all outstanding technical requirements.

Kind regards,

Feng Pan, M.D., Ph.D.

Academic Editor

PLOS ONE

**Comments to the Author**

1. If the authors have adequately addressed your comments raised in a previous round of review and you feel that this manuscript is now acceptable for publication, you may indicate that here to bypass the “Comments to the Author” section, enter your conflict of interest statement in the “Confidential to Editor” section, and submit your "Accept" recommendation.

Reviewer #1: All comments have been addressed

Reviewer #2: All comments have been addressed

Reviewer #3: All comments have been addressed

2. Is the manuscript technically sound, and do the data support the conclusions?

Reviewer #1: Yes

Reviewer #2: Yes

Reviewer #3: Yes

3. Has the statistical analysis been performed appropriately and rigorously? 

Reviewer #1: Yes

Reviewer #2: Yes

Reviewer #3: Yes

4. Have the authors made all data underlying the findings in their manuscript fully available?

Reviewer #1: Yes

Reviewer #2: Yes

Reviewer #3: Yes

5. Is the manuscript presented in an intelligible fashion and written in standard English?

Reviewer #1: Yes

Reviewer #2: Yes

Reviewer #3: Yes

6. Review Comments to the Author

Reviewer #1: Dear Authors, Thank you for addressing all the suggestions in the review. I have no further comments.

Reviewer #2: You have addressed all of my comments so I am recommending acceptance. Not sure why I have to use so many characters to just sign off on an article, and I think this checklist should be streamlined for when accepting an article.

Reviewer #3: Thank you for addressing my points. It would be great if the data concerning the relative participation per institution could be included prior to publication. That would greatly facilitate to estimate one potential source of sampling bias.

7. PLOS authors have the option to publish the peer review history of their article (what does this mean?). If published, this will include your full peer review and any attached files.

Reviewer #1: No

Reviewer #2: **Yes: **Gary E McPherson

Reviewer #3: No

---

## [Editor Report · Acceptance letter]

16 Nov 2020

PONE-D-20-17856R1 

Prevalence and associated factors of playing-related musculoskeletal disorders among music students in Europe. Baseline findings from the Risk of Music Students (RISMUS) longitudinal multicentre study 

Dear Dr. Cruder:

I'm pleased to inform you that your manuscript has been deemed suitable for publication in PLOS ONE. Congratulations! Your manuscript is now with our production department. 

Kind regards, 

on behalf of

Dr. Feng Pan 

Academic Editor

PLOS ONE